# RETHINKING GOAL-CONDITIONED SUPERVISED LEARNING AND ITS CONNECTION TO OFFLINE RL

**Rui Yang**[1], **Yiming Lu**[1], **Wenzhe Li**[1], **Hao Sun**[2],
**Meng Fang**[3], **Yali Du**[4], **Xiu Li**[1], **Lei Han**[5*], **Chongjie Zhang**[1*]
[1]Tsinghua University, [2]University of Cambridge,
[3]Eindhoven University of Technology, [4]King's College London, [5]Tencent Robotics X
{yangrui19, luym19, lwz21}@mails.tsinghua.edu.cn, hs789@cam.ac.uk
m.fang@tue.nl, yali.du@kcl.ac.uk, li.xiu@sz.tsinghua.edu.cn
lxhan@tencent.com, chongjie@tsinghua.edu.cn

## ABSTRACT

Solving goal-conditioned tasks with sparse rewards using self-supervised learning is promising because of its simplicity and stability over current reinforcement learning (RL) algorithms. A recent work, called Goal-Conditioned Supervised Learning (GCSL), provides a new learning framework by iteratively relabeling and imitating self-generated experiences. In this paper, we revisit the theoretical property of GCSL — optimizing a lower bound of the goal reaching objective, and extend GCSL as a novel offline goal-conditioned RL algorithm. The proposed method is named Weighted GCSL (WGCSL), in which we introduce an advanced compound weight consisting of three parts (1) discounted weight for goal relabeling, (2) goal-conditioned exponential advantage weight, and (3) best-advantage weight. Theoretically, WGCSL is proved to optimize an equivalent lower bound of the goal-conditioned RL objective and generates monotonically improved policies via an iterated scheme. The monotonic property holds for any behavior policies, and therefore WGCSL can be applied to both online and offline settings. To evaluate algorithms in the offline goal-conditioned RL setting, we provide a benchmark including a range of point and simulated robot domains. Experiments in the introduced benchmark demonstrate that WGCSL can consistently outperform GCSL and existing state-of-the-art offline methods in the fully offline goal-conditioned setting.

## 1 INTRODUCTION

Reinforcement learning (RL) enables automatic skill learning and has achieved great success in various tasks (Mnih et al., 2015; Lillicrap et al., 2016; Haarnoja et al., 2018; Vinyals et al., 2019). Recently, goal-conditioned RL is gaining attention from the community, as it encourages agents to reach multiple goals and learn general policies (Schaul et al., 2015). Previous advanced works for goal-conditioned RL consist of a variety of approaches based on hindsight experience replay (Andrychowicz et al., 2017; Fang et al., 2019), exploration (Florensa et al., 2018; Ren et al., 2019; Pitis et al., 2020), and imitation learning (Sun et al., 2019; Ding et al., 2019; Sun et al., 2020; Ghosh et al., 2021). However, these goal-conditioned methods need intense online interaction with the environment, which could be costly and dangerous for real-world applications.

A direct solution to learn goal-conditioned policy from offline data is through imitation learning. For instance, goal-conditioned supervised learning (GCSL) (Ghosh et al., 2021) iteratively relabels collected trajectories and imitates them directly. It is substantially simple and stable, as it does not require expert demonstrations or value estimation. Theoretically, GCSL guarantees to optimize over a lower bound of the objective for goal reaching problem. Although hindsight relabeling (Andrychowicz et al., 2017) with future reached states can be optimal under certain conditions (Eysenbach et al., 2020), it would generate non-optimal experiences in more general offline goal-conditioned RL set-

---

*Corresponding Authors

ting, as discussed in Appendix B.1. As a result, GCSL suffers from the same issue as other behavior cloning methods by assigning uniform weights for all experiences and results in suboptimal policies.

To this end, leveraging the simplicity and stability of GCSL, we generalize it to the offline goal-conditioned RL setting and propose an effective and theoretically grounded method, named Weighted GCSL (WGCSL). We first revisit the theoretical foundation of GCSL by additionally considering discounted rewards, and this allows us to obtain a weighted supervised learning objective. To learn a better policy from the offline dataset and promote learning efficiency, we introduce a more general weighting scheme by considering the importance of different relabeling goals and the expected return estimated with a value function. Theoretically, the discounted weight for relabeling goals contributes to optimizing a tighter lower bound than GCSL. Based on the discounted weight, we show that additionally re-weighting with an exponential function over advantage value guarantees monotonic policy improvement for goal-conditioned RL. Moreover, the introduced weights build a natural connection between WGCSL and offline RL, making WGCSL available for both online and offline setting. Another major challenge in goal-conditioned RL is the multi-modality problem (Lynch et al., 2020), i.e., there are generally many valid trajectories from a state to a goal, which can present multiple counteracting action labels and even impede learning. To tackle the challenge, we further introduce the best-advantage weight under the general weighting scheme, which contributes to the asymptotic performance when the data is multi-modal. Although WGCSL has the cost of learning the value function, we empirically show that it is worth for its remarkable improvement. For the evaluation of offline goal-conditioned RL algorithms, we provide a public benchmark and offline datasets including a set of challenging multi-goal manipulation tasks with a robotics arm or an anthropomorphic hand. Experiments[1] conducted in the introduced benchmark show that WGCSL significantly outperforms other state-of-the-art baselines in the fully offline goal-conditioned settings, especially in the difficult anthropomorphic hand task and when learning from randomly collected datasets with sparse rewards.

## 2 PRELIMINARIES

**Markov Decision Process and offline RL**  The RL problem can be described as a Markov Decision Process (MDP), denoted by a tuple $(\mathcal{S}, \mathcal{A}, \mathcal{P}, r, \gamma)$, where $\mathcal{S}$ and $\mathcal{A}$ are the state and action spaces; $\mathcal{P}$ describes the transition probability as $\mathcal{S} \times \mathcal{A} \times \mathcal{S} \to [0, 1]$; $r : \mathcal{S} \times \mathcal{A} \to \mathbb{R}$ is the reward function and $\gamma \in (0, 1]$ is the discount factor; $\pi : \mathcal{S} \to \mathcal{A}$ denotes the policy, and an optimal policy $\pi^*$ satisfies $\pi^* = \arg\max_\pi \mathbb{E}_{s_1 \sim \rho(s_1), a_t \sim \pi(\cdot|s_t), s_{t+1} \sim \mathcal{P}(\cdot|s_t, a_t)}[\sum_{t=1}^{\infty} \gamma^{t-1} r(s_t, a_t)]$, where $\rho(s_1)$ is the distribution of initial states. For offline RL problems, the agent can only access a static dataset $\mathcal{D}$, and is not allowed to interact with the environment during the training process. The offline data can be collected by some unknown policies.

**Goal-conditioned RL**  Goal-conditioned RL further considers a goal space $\mathcal{G}$. The policy $\pi : \mathcal{S} \times \mathcal{G} \to \mathcal{A}$ and the reward function $r : \mathcal{S} \times \mathcal{G} \times \mathcal{A} \to \mathbb{R}$ are both conditioned on goal $g \in \mathcal{G}$. The agent learns to maximize the expected discounted cumulative return $J(\pi) = \mathbb{E}_{g \sim p(g), s_1 \sim \rho(s_1), a_t \sim \pi(\cdot|s_t), s_{t+1} \sim \mathcal{P}(\cdot|s_t, a_t)}\big[\sum_{t=1}^{\infty} \gamma^{t-1} r(s_t, a_t, g)\big]$, where $p(g)$ is the distribution over goals $g$. We study goal-conditioned RL with sparse rewards, and the reward function is typically defined as:

$$r(s_t, a_t, g) = \begin{cases} 1, & ||\phi(s_t) - g||_2^2 < \text{some threshold} \\ 0, & \text{otherwise} \end{cases},$$

where $\phi : \mathcal{S} \to \mathcal{G}$ is a mapping from states to goals. For theoretical analysis in Section 3, we consider MDPs with discrete state/goal space, such that the reward function can be written as $r(s_t, a_t, g) = 1[\phi(s_t) = g]$, which provides a positive signal only when $s_t$ achieves $g$. The value function $V : \mathcal{S} \times \mathcal{G} \to \mathbb{R}$ is defined as $V^\pi(s, g) = \mathbb{E}_{a_t \sim \pi, s_{t+1} \sim \mathcal{P}(\cdot|s_t, a_t)}\big[\sum_{t=1}^{\infty} \gamma^{t-1} r(s_t, a_t, g)|s_1 = s\big]$.

**Goal-conditioned Supervised Learning**  Different with goal-conditioned RL that maximizes discounted cumulative return, GCSL considers the goal-reaching problem, i.e., maximizing the last-step reward $\mathbb{E}_{g \sim p(g), \tau \sim \pi}\big[r(s_T, a_T, g)\big]$ for trajectories $\tau$ of horizon $T$. GCSL iterates between two process, relabeling and imitating. Trajectories in the replay buffer are relabeled with

---

[1]Code and offline dataset are available at https://github.com/YangRui2015/AWGCSL

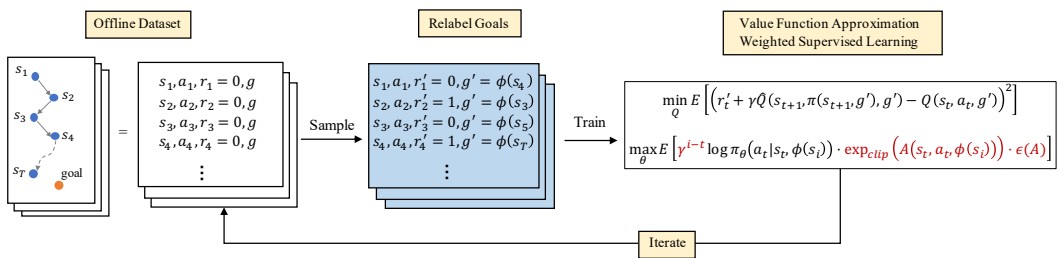

Figure 1: Diagram of WGCSL in the offline goal-conditioned RL setting. WGCSL samples data from the offline dataset and relabels them with hindsight goals. After that, WGCSL learns a value function and updates the policy via a weighted supervised scheme.

hindsight methods (Kaelbling, 1993; Andrychowicz et al., 2017) to form a relabeled dataset $D_{relabel} = \{(s_t, a_t, g')\}$, where $g' = \phi(s_i)$ for $i \geq t$ denotes the relabeled goal. The policy is optimized through supervised learning to mimic those relabeled transitions: $\theta = \arg\max_\theta \mathbb{E}_{(s_t, a_t, g') \sim D_{relabel}} \log \pi_\theta(a_t | s_t, g')$.

## 3 WEIGHTED GOAL-CONDITIONED SUPERVISED LEARNING

In this section, we will revisit GCSL and introduce the general weighting scheme which generalizes GCSL to offline goal-conditioned RL.

### 3.1 REVISITING GOAL-CONDITIONED SUPERVISED LEARNING

As an imitation learning method, GCSL sticks the agent's policy to the relabeled data distribution, therefore it naturally alleviates the problem of out-of-distribution actions. Besides, it has the potential to reach any goal in the offline dataset with hindsight relabeling and the generalization ability of neural networks. Despite its advantages, GCSL has a major disadvantage for offline goal-conditioned RL, i.e., it only considers the last step reward $r(s_T, a_T, g)$ and generally results in suboptimal policies.

**A Motivating Example** We provide training results of GCSL in the PointReach task in Figure 2. The objective of the task is to move a point from the starting position to the desired goal as quickly as possible. The offline dataset is collected using a random policy. As shown in Figure 2, GCSL learns a suboptimal policy which detours to reach goals. This is because GCSL only considers the last step reward $r(s_T, a_T, g)$. As long as the trajectories reaching the goal at the end of the episode, they are all considered equally to GCSL. To improve the learned policy, a straightforward way is to evaluate the importance of samples or trajectories for policy learning using importance weights. As a comparison, Weighted GCSL (WGCSL), which uses a novel weighting scheme, learns the optimal policy in the PointReach task. We will derive the formulation of WGCSL in the following analysis.

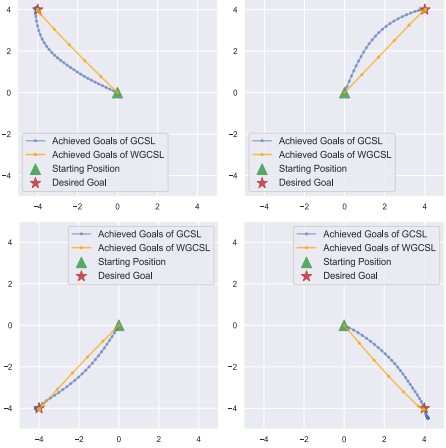

Figure 2: Visualization of the trajectories generated by GCSL (blue) and WGCSL (orange) in the PointReach task.

**Connection between Goal-Conditioned RL and SL** GCSL has been proved to optimize a lower bound on the goal-reaching objective (Ghosh et al., 2021), which is different from the goal-conditioned RL objective that optimizes the cumulative return. In this section, we will introduce a surrogate function for goal-conditioned RL and then derive the connection between goal-conditioned RL and weighted goal-conditioned supervised learning (WGCSL). For

overall consistency, we provide the formulation of WGCSL as below:

$$J_{WGCSL}(\pi) = \mathbb{E}_{g \sim p(g), \tau \sim \pi_b(\cdot|g), t \sim [1,T], i \sim [t,T]} \left[ w_{t,i} \log \pi_\theta(a_t|s_t, \phi(s_i)) \right], \tag{1}$$

where $p(g)$ is the distribution of desired goals and $\pi_b$ refers to the data collecting policy. The weight $w_{t,i}$ is subscripted by both the time step $t$ and the index $i$ of the relabeled goal. When $w_{t,i} = 1$, $J_{WGCSL}$ reduces to GCSL (Ghosh et al., 2021), and we denote the corresponding objective as $J_{GCSL}$ for comparison convenience. Note that in Eq. 1, $\phi(s_i), i \geq t$ implies that any future visited states after $s_t$ can be relabeled as goals.

**Theorem 1.** *Assume a finite-horizon discrete MDP, a stochastic discrete policy $\pi$ which selects actions with non-zero probability and a sparse reward function $r(s_t, a_t, g) = 1[\phi(s_t) = g]$, where $\phi$ is the state-to-goal mapping and $1[\phi(s_t) = g]$ is an indicator function. Given trajectories $\tau = (s_1, a_1, \cdots, s_T, a_T)$ and discount factor $\gamma \in (0, 1]$, let the weight $w_{t,i} = \gamma^{i-t}, t \in [1,T], i \in [t,T]$, then the following bounds hold:*

$$J_{surr}(\pi) \geq T \cdot J_{WGCSL}(\pi) \geq T \cdot J_{GCSL}(\pi),$$

*where $J_{surr}(\pi) = \frac{1}{T} \mathbb{E}_{g \sim p(g), \tau \sim \pi_b(\cdot|g)} \left[ \sum_{t=1}^T \log \pi(a_t|s_t, g) \sum_{i=t}^T \gamma^{i-1} \cdot 1[\phi(s_i) = g] \right]$ is a surrogate function of $J(\pi)$.*

We defer the proof to Appendix B.2, where we also show that under mild conditions, (1) $J_{surr}$ is a lower bound of $\log J$, and (2) $J_{surr}$ shares the same gradient direction with $J$ at $\pi_b$. Theorem 1 reveals the connection between the goal-conditioned RL objective and the WGCSL/GCSL objective. Meanwhile, it suggests that GCSL with the discount weight $\gamma^{i-t}$ is a tighter lower bound compared to the unweighted version. We name this weight as *discounted relabeling weight* (DRW) as it intuitively assigns smaller weights on longer trajectories reaching the same relabeled goal.

### 3.2 A More General Weighting Scheme

DRW can be viewed as a weight evaluating the importance of relabeled goals. In this subsection, we consider a more general weighting scheme using a weight function measuring the importance of state-action-goal combination, revealed by the following corollary.

**Corollary 1.** *Suppose function $h(s, a, g) \geq 1$ over the state-action-goal combination. Given trajectory $\tau = (s_1, a_1, \cdots, s_T, a_T)$, let $w_{t,i} = \gamma^{i-t} h(s_t, a_t, \phi(s_i)), t \in [1,T], i \in [t,T]$, then the following bound holds:*

$$J_{surr}(\pi) \geq T \cdot J_{WGCSL}(\pi).$$

The proof are provided in Appendix B.3. Naturally, the Q-value in RL with some constant shift is a candidate of the function family $h(s, a, g)$. Inspired by prior offline RL approaches (Wang et al., 2018; Peng et al., 2019; Nair et al., 2020), we choose $h(s, a, g) = \exp(A(s, a, g) + C)$, an exponential function over the goal-conditioned advantage, which we refer to as *goal-conditioned exponential advantage weight* (GEAW). The constant $C$ is used to ensure that $h(s, a, g) \geq 1$. The intuition is that GEAW assigns larger weight for samples with higher values. In addition, the exponential advantage weighted formulation has been demonstrated to be a closed-form solution of an offline RL problem, where the learned policy is constrained to stay close to the behavior policy (Wang et al., 2018). Compared to DRW, GEAW evaluates the importance of state-action-goal samples, taking advantage of the universal value function (Schaul et al., 2015).

To tackle the multi-modality challenge in goal-conditioned RL (Lynch et al., 2020), we further introduce the *best-advantage weight* (BAW) based on the learned value function. BAW has the following form:

$$\epsilon(A(s_t, a_t, \phi(s_i))) = \begin{cases} 1, & A(s_t, a_t, \phi(s_i)) > \hat{A} \\ \epsilon_{min}, & \text{otherwise} \end{cases} \tag{2}$$

where $\hat{A}$ is a threshold, and $\epsilon_{min}$ is a small positive value. Implementation details of BAW can be found in Section 4. In our implementation, $\hat{A}$ gradually increases. Therefore, BAW gradually leads the policy toward the modal with the highest return rather than sticks to a position between multiple modals, especially when fitting using a Gaussian policy. Combining all three introduced weights, we have the weighting form: $w_{t,i} = \gamma^{i-t} \exp(A(s_t, a_t, \phi(s_i)) + C) \cdot \epsilon(A(s_t, a_t, \phi(s_i)))$. The constant $C$ only introduces a coefficient independent of $t, i$ for all transitions, therefore, we simply omit it for further analysis.

### 3.3 POLICY IMPROVEMENT VIA WGCSL

We formally show that combining the three introduced weights, WGCSL can consistently improve the policy learned from the offline dataset. First, we assume there exists a policy $\pi_{relabel}$ that can produce the relabeled experiences $D_{relabel}$. GCSL is essentially equivalent to imitating $\pi_{relabel}$:

$$\arg\min_\theta D_{KL}(\pi_{relabel}||\pi_\theta) = \arg\max_\theta \mathbb{E}_{D_{relabel}}[\log \pi_\theta(a_t|s_t, \phi(s_i)))].$$

Following (Wang et al., 2018), solving WGCSL is equivalent to imitating a new goal-conditioned policy $\tilde{\pi}$ defined as:

$$\tilde{\pi}(a_t|s_t, \phi(s_i)) = \gamma^{i-t}\pi_{relabel}(a_t|s_t, \phi(s_i)) \cdot \exp(A(s_t, a_t, \phi(s_i))) \cdot \epsilon(A(s_t, a_t, \phi(s_i))),$$

Then, we have the following proposition:

**Proposition 1.** *Under certain conditions, $\tilde{\pi}$ is uniformly as good as or better than $\pi_{relabel}$. That is, $\forall s_t, \phi(s_i) \in D_{relabel}, i \geq t, V^{\tilde{\pi}}(s_t, \phi(s_i)) \geq V^{\pi_{relabel}}(s_t, \phi(s_i))$, where $D_{relabel}$ contains the experiences after goal relabeling.*

The proof can be found in Appendix B.4. Proposition 1 implies that WGCSL can generate a uniformly non-worse policy using the relabeled data $D_{relabel}$. To be rigorous, we also need to discuss the relationship between the relabeled return and the original return. In fact, there is a monotonic improvement for the relabeled return over the original one when relabeling with inverse RL (Eysenbach et al., 2020). Below, we provide a simpler relabeling strategy which also offers monotonic value improvement.

**Proposition 2.** *Assume the state space can be perfectly mapped to the goal space, $\forall s_t \in S, \phi(s_t) \in G$, and the dataset has sufficient coverage. We define a relabeling strategy for $(s_t, g)$ as*

$$g' = \begin{cases} \phi(s_i), i \sim U[t, T], & if\ \phi(s_i) \neq g, \forall i \geq t, \\ g, & otherwise. \end{cases}$$

*We also assume that when $g' \neq g$, the strategy only accepts relabeled trajectories with higher future returns than any trajectory with goal $g'$ in the original dataset $D$. Then, $\pi_{relabel}$ is uniformly as good as or better than $\pi_b$, i.e., $\forall s_t, g, \in D, V^{\pi_{relabel}}(s_t, g) \geq V^{\pi_b}(s_t, g)$, where $\pi_b$ is the behavior policy forming the dataset $D$.*

The proof is provided in Appendix B.5. In Proposition 2, the relabeling strategy is slightly different from the strategy of relabeling with random future states (Andrychowicz et al., 2017). When there is no successful future states, the relabeling strategy in Proposition 2 randomly samples from the future visited states for relabeling, otherwise it keeps the original goal. For random datasets, there are rare successful states and the relabeling strategy in Proposition 2 acts similarly to relabeling with random future states. Combining Propositions 1 and 2, we know that under certain assumptions, after performing relabeling and weighted supervised learning, the policy can be monotonically improved. In fully offline settings, WGCSL is able to learn a better policy than the policy learned by GCSL and the behavior policy generating the offline dataset.

## 4 ALGORITHM

In this section, we summarize our proposed algorithm. Denote the relabeled dataset as $D_{relabel} = (s_t, a_t, r'_t, g'), t \in [1, T], g' = \phi(s_i), r'_t = r(s_t, a_t, \phi(s_i)), i \geq t$, and we maximize the following WGCSL objective based on the relabeled data

$$\hat{J}_{WGCSL}(\pi) = \mathbb{E}_{(s_t, a_t, \phi(s_i))\sim\mathcal{D}relabel}[w_{t,i} \cdot \log \pi_\theta(a_t|s_t, \phi(s_i))], \tag{3}$$

where the weight $w_{t,i}$ is composed of three parts conforming to the general weighting scheme

$$w_{t,i} = \gamma^{i-t} \cdot \exp_{clip}(A(s_t, a_t, \phi(s_i))) \cdot \epsilon(A(s_t, a_t, \phi(s_i))) \tag{4}$$

(1) $\gamma^{i-t}$ is the discounted relabeling weight (DRW) as introduced in Section 3.1.

(2) $\exp_{clip}(A(s_t, a_t, \phi(s_i))) = \mathrm{clip}(\exp(A(s_t, a_t, \phi(s_i))), 0, M)$ is the goal-conditioned exponential advantage weight (GEAW) with a clipped range $(0, M]$, where $M$ is some positive number for numerical stability. In our experiments, $M$ is set as 10 for all the tasks.

Figure 3: Goal-conditioned tasks: (a) PointReach, (b) PointRooms, (c) Reacher, (d) SawyerReach, (e) SawyerDoor, (f) FetchReach, (g) FetchPush, (h) FetchSlide, (i) FetchPick, (j) HandReach.

(3) $\epsilon(A(s_t, a_t, \phi(s_i)))$ is the best-advantage weight (BAW) with the form in Eq. 2. In our experiments, $\hat{A}$ is set as $N$ percentile of recent advantage values and $\epsilon_{min}$ is set as 0.05. Motivated by curriculum learning (Bengio et al., 2009), $N$ gradually increases from 0 to 80, i.e., we learn from all the experiences in the beginning stage and converge to learning from samples with the top 20% advantage values.

The advantage value in Eq. 3 can be estimated by $A(s_t, a_t, g') = r(s_t, a_t, g') + \gamma V(s_{t+1}, g') - V(s_t, g')$, and we learn a Q-value function $Q(s_t, a_t, g')$ by minimizing the TD error

$$\mathcal{L}_{TD} = \mathbb{E}_{(s_t, a_t, g', r'_t, s_{t+1}) \sim D_{relabel}}[(r'_t + \gamma \hat{Q}(s_{t+1}, \pi(s_{t+1}, g'), g')) - Q(s_t, a_t, g'))^2],$$

where we relabel original goals $g$ as $g' = \phi(s_i)$ with a probability of 80% and remain the original goal unchanged for the rest 20% time. For simplicity, in our experiments, once it falls into the 80% probability of goal relabeling, we use purely relabeling with random future states, no matter whether there exist successful future states in the trajectory or not, to avoid parsing all the future states as discussed in Appendix B.5. $\hat{Q}$ refers to the target network which is slowly updated to stabilize training. The same value function training method is adopted in (Andrychowicz et al., 2017; Fang et al., 2019). The value function $V(s_t, g')$ is estimated using Q-value function as $V(s_t, g') = Q(s_t, \pi(s_{t+1}, g'), g')$. The overall framework of WGCSL is presented in Figure 1 and the entire algorithm is provided in Appendix C.

## 5 EXPERIMENTS

In our experiments, we demonstrate our proposed method on a diverse collection of continuous sparse-reward goal-conditioned tasks.

### 5.1 ENVIRONMENTS AND EXPERIMENTAL SETTINGS

As shown in Figure 3, the introduced benchmark includes two point-based environments, and eight simulated robot environments. In all tasks, the rewards are sparse and binary: the agent receives a reward of $+1$ if it achieves a desired goal and a reward of $0$ otherwise. More details about those environments are provided in Appendix F. For the evaluation of offline goal-conditioned algorithms, we collect two types of offline dataset, namely 'random' and 'expert', following prior offline RL work (Fu et al., 2020). Each dataset has the sample size of $2 \times 10^6$ for hard tasks (FetchPush, FetchSlide, FetchPick and HandReach) and $1 \times 10^5$ for other relatively easy tasks. The 'random' dataset is collected by a uniform random policy, while the 'expert' dataset is collected by the final policy trained using online HER. We also add Gaussian noise with zero mean and 0.2 standard deviation to increase the diversity of the 'expert' dataset. This may lead to the consequence that the deterministic policy learned by behavior cloning achieves a higher average return than that in the dataset, which has also been observed in previous works (Kumar et al., 2019; Fujimoto et al., 2019). More details about the two datasets are provided in Appendix D. Beyond the offline setting, we also evaluate our method in the online setting and the results are reported in Appendix E.8.

As for the implementation of GCSL and WGCSL, we utilize the Diagonal Gaussian policy with a mean vector $\pi_\theta(s_t, g)$ and a constant variance $\sigma^2$ for continuous control. In the offline setting, $\sigma$ does not need to be defined explicitly because we evaluate the policy without randomness. The loss function of GCSL can be rewritten as: $\mathcal{L}_{GCSL} = \mathbb{E}_{(s_t, a_t, \phi(s_i)) \sim D_{relabel}}[\|\pi_\theta(s_t, \phi(s_i)) - a_t\|_2^2]$. Our implemented WGCSL learns a policy to minimize the following loss

$$\mathcal{L}_{WGCSL} = \mathbb{E}_{(s_t, a_t, \phi(s_i)) \sim D_{relabel}}[w_{t,i} \cdot \|\pi_\theta(s_t, \phi(s_i)) - a_t\|_2^2].$$

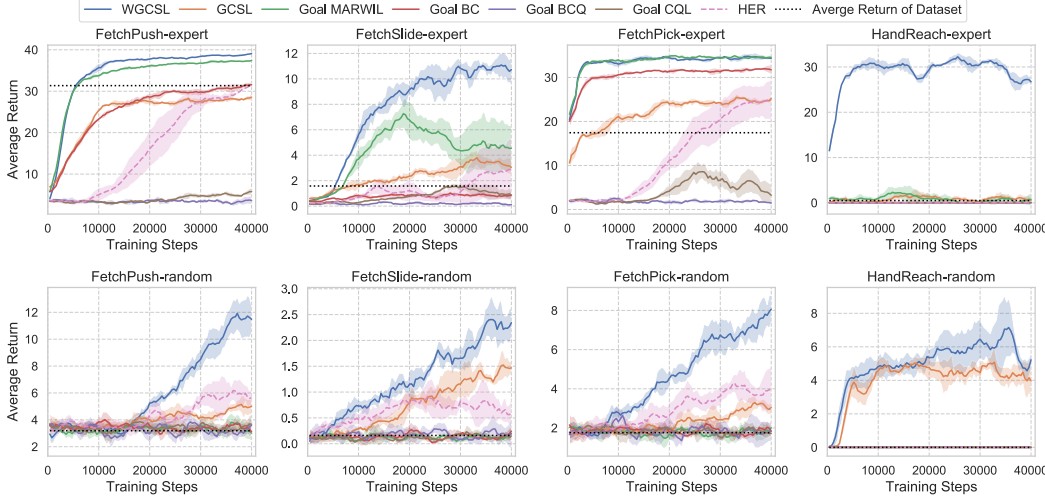

Figure 4: Performance on expert (top row) and random (bottom row) offline dataset of four simulated manipulation tasks. Results are averaged over 5 random seeds and the shaded region represents the standard deviation.

Table 1: Average return of algorithms on the offline goal-conditioned benchmark. g-MARWIL, g-BC, g-BCQ and g-CQL, are short for Goal MARWIL, Goal BC, Goal BCQ, Goal CQL, respectively. The suffix "-e" and "-r" are short for expert and random datasets, respectively.

| Task Name | WGCSL | GCSL | g-MARWIL | g-BC | g-BCQ | g-CQL | HER |
|---|---|---|---|---|---|---|---|
| PointReach-e | **44.40** $\pm 0.14$ | 39.27 $\pm 0.48$ | **42.95** $\pm 0.15$ | 39.36 $\pm 0.48$ | 40.42 $\pm 0.57$ | 39.75 $\pm 0.33$ | 24.68 $\pm 7.07$ |
| PointRooms-e | **36.15** $\pm 0.85$ | 33.05 $\pm 0.54$ | **36.02** $\pm 0.57$ | 33.17 $\pm 0.52$ | 32.37 $\pm 1.77$ | 30.05 $\pm 0.38$ | 12.41 $\pm 8.59$ |
| Reacher-e | **40.57** $\pm 0.20$ | 36.42 $\pm 0.30$ | 38.89 $\pm 0.17$ | 35.72 $\pm 0.37$ | 39.57 $\pm 0.08$ | **42.23** $\pm 0.12$ | 8.27 $\pm 4.33$ |
| SawyerReach-e | **40.12** $\pm 0.29$ | 33.65 $\pm 0.38$ | 37.42 $\pm 0.31$ | 32.91 $\pm 0.31$ | **39.49** $\pm 0.33$ | 19.33 $\pm 0.45$ | 26.48 $\pm 6.23$ |
| SawyerDoor-e | 42.81 $\pm 0.23$ | 35.67 $\pm 0.09$ | 40.03 $\pm 0.16$ | 35.03 $\pm 0.20$ | 40.13 $\pm 0.75$ | **45.86** $\pm 0.11$ | 44.09 $\pm 0.65$ |
| FetchReach-e | 46.33 $\pm 0.04$ | 41.72 $\pm 0.31$ | **45.01** $\pm 0.11$ | 42.03 $\pm 0.25$ | 35.18 $\pm 3.09$ | 1.03 $\pm 0.26$ | **46.73** $\pm 0.14$ |
| FetchPush-e | **39.11** $\pm 0.17$ | 28.56 $\pm 0.96$ | 37.42 $\pm 0.22$ | 31.56 $\pm 0.61$ | 3.62 $\pm 0.96$ | 5.76 $\pm 0.83$ | 31.53 $\pm 0.47$ |
| FetchSlide-e | **10.73** $\pm 1.09$ | 3.05 $\pm 0.62$ | 4.55 $\pm 1.79$ | 0.84 $\pm 0.35$ | 0.12 $\pm 0.10$ | 0.86 $\pm 0.38$ | 2.86 $\pm 2.40$ |
| FetchPick-e | 34.37 $\pm 0.51$ | 25.22 $\pm 0.85$ | **34.56** $\pm 0.54$ | 31.75 $\pm 1.19$ | 1.46 $\pm 0.29$ | 3.23 $\pm 2.52$ | 24.79 $\pm 4.49$ |
| HandReach-e | **26.73** $\pm 1.20$ | 0.57 $\pm 0.68$ | 0.81 $\pm 1.59$ | 0.06 $\pm 0.03$ | 0.04 $\pm 0.04$ | 0.00 $\pm 0.00$ | 0.05 $\pm 0.07$ |
| PointReach-r | 44.30 $\pm 0.24$ | 30.80 $\pm 1.74$ | 7.67 $\pm 1.97$ | 1.37 $\pm 0.09$ | 1.78 $\pm 0.14$ | 1.52 $\pm 0.26$ | **45.17** $\pm 0.13$ |
| PointRooms-r | 35.52 $\pm 0.80$ | 24.10 $\pm 0.81$ | 4.67 $\pm 0.80$ | 1.43 $\pm 0.18$ | 1.61 $\pm 0.17$ | 1.29 $\pm 0.37$ | **36.16** $\pm 1.16$ |
| Reacher-r | **41.12** $\pm 0.11$ | 22.52 $\pm 0.77$ | 15.35 $\pm 1.95$ | 1.66 $\pm 0.30$ | 2.52 $\pm 0.28$ | 2.54 $\pm 0.17$ | 34.48 $\pm 8.12$ |
| SawyerReach-r | **41.05** $\pm 0.19$ | 14.86 $\pm 3.27$ | 11.30 $\pm 2.12$ | 0.58 $\pm 0.21$ | 1.36 $\pm 0.14$ | 1.18 $\pm 0.29$ | 39.27 $\pm 2.16$ |
| SawyerDoor-r | **36.82** $\pm 3.20$ | 25.86 $\pm 1.12$ | 25.33 $\pm 1.46$ | 3.73 $\pm 0.83$ | 9.82 $\pm 1.08$ | 4.36 $\pm 0.86$ | 28.85 $\pm 1.99$ |
| FetchReach-r | 46.50 $\pm 0.09$ | 38.26 $\pm 0.24$ | 30.86 $\pm 8.49$ | 0.84 $\pm 0.31$ | 0.19 $\pm 0.04$ | 0.97 $\pm 0.23$ | **47.01** $\pm 0.07$ |
| FetchPush-r | **11.48** $\pm 1.03$ | 5.01 $\pm 0.64$ | 3.01 $\pm 0.71$ | 3.14 $\pm 0.25$ | 3.60 $\pm 0.42$ | 3.67 $\pm 0.65$ | 5.65 $\pm 0.64$ |
| FetchSlide-r | **2.34** $\pm 0.28$ | **1.47** $\pm 0.12$ | 0.19 $\pm 0.11$ | 0.25 $\pm 0.13$ | 0.20 $\pm 0.29$ | 0.15 $\pm 0.09$ | 0.59 $\pm 0.30$ |
| FetchPick-r | **8.06** $\pm 0.71$ | 3.05 $\pm 0.42$ | 2.01 $\pm 0.46$ | 1.84 $\pm 0.17$ | 1.84 $\pm 0.58$ | 1.73 $\pm 0.17$ | 3.91 $\pm 1.29$ |
| HandReach-r | **5.23** $\pm 0.55$ | **3.96** $\pm 0.81$ | 0.00 $\pm 0.00$ | 0.00 $\pm 0.00$ | 0.00 $\pm 0.00$ | 0.00 $\pm 0.00$ | 0.00 $\pm 0.01$ |

where $w_{t,i}$ is defined in Eq 4. The policy networks of GCSL and WGCSL are 3-layer MLP with 256 units each layer and relu activation. Besides, WGCSL learns a value network with the same structure except for the input and output layers. The batch size is set as 128 for the first 6 tasks, and 512 for 4 harder tasks from FetchPush to HandReach. We use Adam optimizer with a learning rate of $5 \times 10^{-4}$. For all the experiments we repeat for 5 different random seeds and report the average evaluation performance with standard deviation.

## 5.2 EXPERIMENTAL RESULTS

In the offline goal-conditioned setting, we compare WGCSL with **GCSL**, **HER** and modified versions of most related offline works such as goal-conditioned MARWIL (Wang et al.,

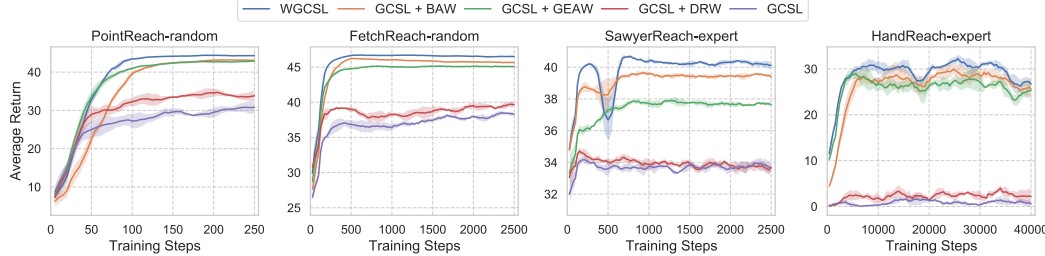

Figure 5: Ablation studies of WGCSL in the offline setting.

2018) (**Goal MARWIL**), goal-conditioned behavior cloning (Bain & Sammut, 1995) (**Goal BC**), goal-conditioned Batch-Constrained Q-Learning (Fujimoto et al., 2019) (**Goal BCQ**), and goal-conditioned Conservative Q-learning (Kumar et al., 2020) (**Goal CQL**). The results are presented in Table 1 and Figure 4. Additional comparison results with more baselines and results on 10% dataset are provided in Appendix E.

**Expert Datasets**   The top row of Figure 4 reports the performance of different algorithms using the expert datasets of four hard tasks. In all these tasks, WGCSL outperforms other baselines in terms of both learning efficiency and final performance. GCSL converges very slowly but can finally reach beyond the average return of expert dataset in FetchSlide and FetchPick tasks. Goal MARWIL is more efficient than GCSL and achieves the second best performance. However, none of the baseline algorithms succeeds in the HandReach task, which might be due to high-dimensional states, goals, actions and the noise used to collect the dataset. In addition, HER itself has the ability to handle offline dataset as the relabeled data is far off-policy (Plappert et al., 2018). But we can conclude from Table 1 that the performance of HER is unstable and inconsistent across different tasks. In contrast, learning curves of WGCSL are consistently stable. It is also verified that techniques used in WGCSL are effective to generate better policies than original and relabeled datasets in offline goal-conditioned RL setting.

**Random Datasets**   The bottom row of Figure 4 shows the training results of different methods on random datasets. We can observe that WGCSL outperforms other baselines by a large margin. Baselines such as Goal MARWIL, Goal BC, Goal BCQ and Goal CQL can hardly learn to reach goals with the random datasets. GCSL and HER can learn relatively good policies in some tasks, which proves the importance of goal relabeling. Interestingly, in tasks such as Reacher, SawyerReach and FetchReach, results of WGCSL and HER on the random dataset exceeds that of the expert dataset. The possible reason is that the data coverage of random dataset could be larger than the expert dataset in these tasks, and the agent can learn to reach more goals via goal relabeling.

**Ablation Studies**   We also include ablation studies for offline experiments in Figure 5. In our ablation studies, we investigate the effect of the three proposed weights, i.e., DRW, GEAW and BAW, by applying them to GCSL respectively. The results demonstrate that all three weights are effective compared to the plain GCSL, while GEAW and BAW show larger benefits. This is reasonable since GEAW and BAW learn additional value networks while DRW does not. Moreover, the results suggest that the learned policy can be improved by combining these weights.

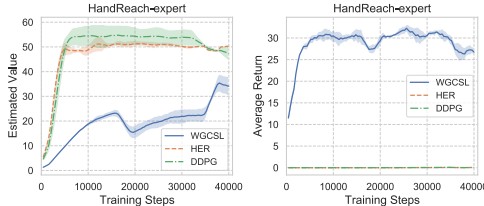

Figure 6: Average estimated value (left) and performance (right) of WGCSL, HER and DDPG. The TD target is clipped to $[0, 50]$, thus the maximum estimated value is around 50.

**Value Estimation**   Off-policy RL algorithms are prone to distributional shift and suffer from value overestimation problem on out-of-distribution actions. Such problem can be exacerbated in tasks with high-dimensional states and actions. As shown in Figure 6, DDPG and HER exhibit large estimated values during offline training. In contrast, WGCSL has a more robust value approximation through weighted supervised learning, thus achieving higher returns in the difficult HandReach task.

## 6    RELATED WORK

Solving goal-conditioned RL task is an important yet challenging domain in reinforcement learning. In those tasks, agents are required to achieve multiple goals rather than a single task (Schaul et al., 2015). Such setting puts much burden on the generalization ability of the learned policy when dealing with unseen goals. Another challenge arises in goal-conditioned RL is the sparse reward (Plappert et al., 2018). HER (Andrychowicz et al., 2017) tackles the sparse reward issue via relabeling the failed rollouts as successful ones to increase the proportion of successful trails during learning. HER is then extended to deal with demonstration (Nair et al., 2018; Ding et al., 2019) and dynamic goals (Fang et al., 2018), and combined with curriculum learning for exploration-exploitation trade-off (Fang et al., 2019). Eysenbach et al. (2020) and Li et al. (2020) further improved the goal sampling strategy from the perspective of inverse RL. Zhao et al. (2021) encouraged agents to control their states to reach goals by optimizing a mutual information objective without external rewards. Different from prior works, we consider the offline setting, where the agent cannot access the environment to collect data in the training phase.

Our work has strong connections to imitation learning (IL). Behavior cloning is one of the simplest IL methods that learns a policy predicting the expert actions (Bain & Sammut, 1995; Bojarski et al., 2016). Inverse RL (Ng et al., 2000) extracts a reward function from demonstrations and then uses the reward function to train policies. Several works also leverage self-imitation to improve the stability and efficiency of RL without expert experience. Self-imitation learning (Oh et al., 2018) imitates the agent's past advantageous decisions and indirectly drives exploration. Sun et al. (2019) introduced a dynamic programming method to learn policies progressively via self-imitation. Lynch et al. (2020) employed VAE to jointly learn a latent plan representation and goal-conditioned policy from replay without the need for rewards. Our work is most relevant to GCSL (Ghosh et al., 2021), which iteratively relabels and imitates its own collected experiences. The difference is that we further generalize GCSL to the offline goal-conditioned RL setting via weighted supervised learning.

Our work belongs to the scope of offline RL. Learning policies from static offline datasets is challenging for general off-policy RL algorithms due to error accumulation with distributional shift (Fujimoto et al., 2019; Kumar et al., 2019). To address such issue, a series of algorithms are proposed to constrain the policy updates to avoid excessive deviation from the data distribution (Wang et al., 2018; Fujimoto et al., 2019; Wu et al., 2019; Yang et al., 2021). MARWIL (Wang et al., 2018) can learn a better policy than the behavior policy of the dataset with theoretical guarantees. In addition to policy regularization, other works leverage techniques such as value underestimation (Kumar et al., 2020; Yu et al., 2020), robust value estimation (Agarwal et al., 2020), data augmentation (Sinha & Garg, 2021; Wang et al., 2021), and Expectile $V$-Learning (Ma et al., 2021) to handle distribution shift. As offline RL can easily overfit to one task, generalizing offline RL to unseen tasks can be even more challenging (Li et al., 2021). Li et al. (2019) tackled the challenge by leveraging the triplet loss for robust task inference. Our method has similar exponential advantage weighted form as MARWIL, however, we tackle goal-conditioned tasks and introduces a general weighting scheme for WGCSL. A most recent work (Chebotar et al., 2021) also addresses the offline goal-conditioned problem through underestimating the value of unseen actions. The difference is that WGCSL is a weighted supervised learning method without explicitly value underestimation. Therefore, WGCSL is substantially simpler, more stable, and more applicable for real-world tasks.

## 7    CONCLUSION

In this paper, we have proposed a novel algorithm, Weighted Goal-Conditioned Supervised Learning (WGCSL), to solve offline goal-conditioned RL problems with sparse rewards, by first revisiting GCSL and then generalizing it to offline goal-conditioned RL. We further derive theoretical guarantees to show that WGCSL can generate monotonically improved policies under certain conditions. Experiments on an introduced benchmark demonstrate that WGCSL outperforms current offline goal-conditioned approaches by a great margin in terms of learning efficiency and convergent performance. For future directions, we are interested in building more stable and efficient offline goal-conditioned RL algorithms based on WGCSL for real-world applications.

## ACKNOWLEDGMENTS

This work was partly supported by the Science and Technology Innovation 2030-Key Project under Grant 2021ZD0201404, in part by Science and Technology Innovation 2030 – "New Generation Artificial Intelligence" Major Project (No. 2018AAA0100904) and National Natural Science Foundation of China (No. 20211300509). Part of this work was done when Rui Yang and Hao Sun worked as interns at Tencent Robotics X Lab.

## ETHICS AND REPRODUCIBILITY STATEMENTS

Our work provides a way to learn general policy from offline data. Learning policy for general purpose is one of the fundamental challenges in artificial general intelligence (AGI). The proposed offline goal-conditioned algorithm can provide new insights into training agents with diverse skills, but currently we cannot foresee any negative social impact of our work. The algorithm is evaluated on simulated robot environments, thus our experiments would not suffer from discrimination/bias/fairness concerns. The details of experimental settings and additional results are also provided in the Appendix. All the implementation code and offline dataset are available in our code link: https://github.com/YangRui2015/AWGCSL.

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

## A    A COMPREHENSIVE SUMMARY FOR WGCSL

WGCSL is strongly efficient in the offline goal-conditioned RL due to the following advantages:

(1) WGCSL learns a universal value function (Schaul et al., 2015) which is more generalizable than the value function learned in a single task, thus alleviating the overfitting problem in offline RL.

(2) WGCSL utilizes goal relabeling which contributes to augment huge amount of data ($\binom{T}{2}$ times) for offline training and meanwhile alleviates the sparse reward issue.

(3) WGCSL leverages weighted supervised learning to reweight the action distribution in the dataset, avoiding the out-of-distribution action problem in offline RL.

(4) In the three weights, GEAW keeps the learned policy close to the offline dataset implicitly and BAW handles the multi-modality problem in offline goal-conditioned RL. WGCSL utilizes the three weights to learn a better policy with theoretical guarantees.

## B    THEORETICAL RESULTS

In our analysis, we will assume finite-horizon MDPs with discrete state and goal spaces.

### B.1    NON-OPTIMALITY OF RELABELING STRATEGY

Given a trajectory $\tau = \{s_1, a_1, \ldots, s_T, a_T\}$, we show that under general offline goal-conditioned RL setting with the sparse reward function $r(s_t, a_t) = 1[\phi(s_t) = g]$, relabeling with future states is not optimal. This is a direct result from Eq. 7 of (Eysenbach et al., 2020), which has proven that the optimal relabelling strategy is

$$q(g|\tau) \propto p(g) \exp \big( \sum_{t=1}^{T} \gamma^{t-1} r(s_t, a_t) - \log Z(g) \big) = p(g) \exp \big( \sum_{t=1}^{T} \gamma^{t-1} \cdot 1[\phi(s_t) = g] - \log Z(g) \big).$$

This is different from simply relabeling with the last state as $q(g|\tau) = 1[g = s_T]$ in Section 4.1 of (Eysenbach et al., 2020) and relabeling with future states as $q(g|s_t, a_t) = \frac{1}{(T-t+1)} 1[g = s_i], t \in [1, T], i \in [t, T]$.

### B.2    PROOF OF THEOREM 1

**Theorem.** *Assume a finite-horizon discrete MDP, a stochastic discrete policy $\pi$ which selects actions with non-zero probability and a sparse reward function $r(s_t, a_t, g) = 1[\phi(s_t) = g]$, where $\phi$ is the state-to-goal mapping and $1[\phi(s_t) = g]$ is an indicator function. Given trajectory $\tau = (s_1, a_1, \cdots, s_T, a_T)$ and discount factor $\gamma \in (0, 1]$, let the weight $w_{t,i} = \gamma^{i-t}, t \in [1, T], i \in [t, T]$, then the following bounds hold:*

$$J_{surr}(\pi) \geq T \cdot J_{WGCSL}(\pi) \geq T \cdot J_{GCSL}(\pi),$$

*where*

$$J_{surr}(\pi) = \frac{1}{T} \cdot \mathbb{E}_{g \sim p(g), \tau \sim \pi_b(\cdot|g)} \left[ \sum_{t=1}^{T} \log \pi(a_t|s_t, g) \sum_{i=t}^{T} \gamma^{i-1} \cdot 1[\phi(s_i) = g] \right]$$

*is a surrogate function of $J(\pi)$.*

*Proof.*

$$J_{surr}(\pi) = \frac{1}{T} \cdot \mathbb{E}_{g \sim p(g), \tau \sim \pi_b(\cdot|g)} \left[ \sum_{t=1}^{T} \log \pi(a_t|s_t, g) \sum_{i=t}^{T} \gamma^{i-1} \cdot 1[\phi(s_i) = g] \right]$$

$$\geq \frac{1}{T} \cdot \mathbb{E}_{g \sim p(g), \tau \sim \pi_b(\cdot|g)} \left[ \sum_{t=1}^{T} \log \pi(a_t|s_t, g) \sum_{i=t}^{T} \gamma^{i-t} \cdot 1[\phi(s_i) = g] \right]$$

$$= \mathbb{E}_{g \sim p(g), \tau \sim \pi_b(\cdot|g), t \sim [1,T]} \left[ \sum_{i=t}^{T} \gamma^{i-t} \log \pi(a_t|s_t, \phi(s_i)) \cdot 1[\phi(s_i) = g] \right]$$

$$\geq T \cdot \mathbb{E}_{g \sim p(g), \tau \sim \pi_b(\cdot|g), t \sim [1,T], i \sim [t,T]} \left[ \gamma^{i-t} \log \pi(a_t|s_t, \phi(s_i)) \cdot 1[\phi(s_i) = g] \right]$$

$$\geq T \cdot \mathbb{E}_{g \sim p(g), \tau \sim \pi_b(\cdot|g), t \sim [1,T], i \sim [t,T]} \left[ \gamma^{i-t} \log \pi(a_t|s_t, \phi(s_i)) \right] \triangleq T \cdot J_{WGCSL}(\pi) \quad (5)$$

$$\geq T \cdot \mathbb{E}_{g \sim p(g), \tau \sim \pi_b(\cdot|g), t \sim [1,T], i \sim [t,T]} \left[ \log \pi(a_t|s_t, \phi(s_i)) \right] \triangleq T \cdot J_{GCSL}(\pi).$$

Eq. 5 holds because of the non-positive logarithmic function over $\pi$. Compared to the surrogate function used in Appendix B.1 of (Ghosh et al., 2021), $J_{surr}(\pi)$ further considers the discounted cumulative return. The theorem shows that WGCSL optimizes over a tighter lower bound of $J_{surr}(\pi)$ compared to GCSL.

**Remark.** While Theorem 1 bounds the surrogate function $J_{surr}$ with $J_{WGCSL}$ and $J_{GCSL}$, here we bridge the gap between $J_{surr}$ and the goal-conditioned RL objective $J$.

We first show that $\log J$ is an upper bound of $J_{surr}$ in addition to KL divergence and a constant. Let $R(\tau) = \sum_{i=1}^{T} \gamma^{i-1} \cdot 1[\phi(s_i) = g]$ and $C(g) = \int \pi_b(\tau|g) R(\tau) d\tau$, and assume $C(g) > 0$. Since practically $g$ is sampled from offline dataset, we can find $C_m = \min_g C(g)$. Similarly to (Kober & Peters, 2014), we have following results using Jensen's inequality and absorbing constants into 'constant':

$$\log J(\pi) = \log \mathbb{E}_{g \sim p(g)} \left[ \int \pi(\tau|g) R(\tau) d\tau \right]$$

$$\geq \mathbb{E}_{g \sim p(g)} \left[ \log C(g) \cdot \int \frac{\pi_b(\tau|g)}{C(g) \cdot \pi_b(\tau|g)} \pi(\tau|g) R(\tau) d\tau \right]$$

$$\geq \mathbb{E}_{g \sim p(g)} \left[ \int \frac{\pi_b(\tau|g) R(\tau)}{C(g)} (\log \pi(\tau|g) - \log \pi_b(\tau|g)) d\tau \right] + \text{constant}$$

$$\geq \mathbb{E}_{g \sim p(g), \tau \sim \pi_b(\cdot|g)} \left[ \frac{1}{C(g)} \log \pi(\tau|g) \sum_{i=1}^{T} \gamma^{i-1} \cdot 1[\phi(s_i) = g] \right] + \text{constant}$$

$$\geq \frac{1}{C_m} \cdot \mathbb{E}_{g \sim p(g), \tau \sim \pi_b(\cdot|g)} \left[ \sum_{t=1}^{T} \log \pi(a_t|s_t, g) \sum_{i=1}^{T} \gamma^{i-1} \cdot 1[\phi(s_i) = g] \right] + \text{constant}$$

$$= \frac{T}{C_m} \cdot J_{surr}(\pi) + \frac{1}{C_m} \cdot \mathbb{E}_{g \sim p(g), \tau \sim \pi_b(\cdot|g)} \left[ \sum_{t=1}^{T} \log \pi(a_t|s_t, g) \sum_{i=1}^{t-1} \gamma^{i-1} \cdot 1[\phi(s_i) = g] \right] + \text{constant}$$

$$\geq \frac{T}{C_m} \cdot J_{surr}(\pi) + \frac{T}{C_m} \cdot \mathbb{E}_{g \sim p(g), \tau \sim \pi_b(\cdot|g)} \left[ \sum_{t=1}^{T} \log \pi(a_t|s_t, g) \right] + \text{constant}$$

$$= \frac{T}{C_m} \cdot J_{surr}(\pi) + \frac{T^2}{C_m} \cdot \mathbb{E}_{g \sim p(g), (s_t, a_t) \sim \pi_b(\cdot|g)} \left[ \log \pi(a_t|s_t, g) \right] + \text{constant}$$

$$= \frac{T}{C_m} \cdot J_{surr}(\pi) + \frac{T^2}{C_m} \cdot \mathbb{E}_{g \sim p(g), (s_t, a_t) \sim \pi_b(\cdot|g)} \left[ \log \pi(a_t|s_t, g) - \log \pi_b(a_t|s_t, g) \right] + \text{constant}$$

$$= \frac{T}{C_m} \cdot J_{surr}(\pi) - \frac{T^2}{C_m} \cdot \mathbb{E}_{g \sim p(g), s_t \sim \pi_b(\cdot|g)} \left[ D_{KL}(\pi_b(\cdot|s_t, g) \| \pi(\cdot|s_t, g)) \right] + \text{constant}.$$

Naturally, in offline setting, we explicitly or implicitly bound the KL divergence between $\pi_b$ and $\pi$ via policy constraints or (weighted) supervised learning. Therefore, we can regard $J_{surr}$ as a lower bound of $\log J$.

Furthermore, We show that $J_{surr}(\pi)$ shares the same gradient with $J(\pi)$ when $\pi = \pi_b$ after scaling with $\frac{1}{T}$.

$$J(\pi) = \mathbb{E}_{g \sim p(g), \tau \sim \pi(\cdot|g)} \left[ \sum_{i=1}^{T} \gamma^{i-1} \cdot 1[\phi(s_i) = g] \right],$$

$$\nabla J(\pi_b) = \mathbb{E}_{g \sim p(g)} \left[ \int_{\tau} \nabla \pi_b(\tau|g) \sum_{i=1}^{T} \gamma^{i-1} \cdot 1[\phi(s_i) = g] d\tau \right]$$

$$= \mathbb{E}_{g \sim p(g)} \left[ \int_{\tau} \pi_b(\tau|g) \nabla \log \pi_b(\tau|g) \sum_{i=1}^{T} \gamma^{i-1} \cdot 1[\phi(s_i) = g] d\tau \right]$$

$$= \mathbb{E}_{g \sim p(g), \tau \sim \pi_b(\cdot|g)} \left[ \sum_{t=1}^{T} \nabla \log \pi_b(a_t|s_t, g) \sum_{i=1}^{T} \gamma^{i-1} \cdot 1[\phi(s_i) = g] \right]$$

$$= \mathbb{E}_{g \sim p(g), \tau \sim \pi_b(\cdot|g)} \left[ \sum_{t=1}^{T} \nabla \log \pi_b(a_t|s_t, g) \sum_{i=t}^{T} \gamma^{i-1} \cdot 1[\phi(s_i) = g] \right]$$

$$= T \cdot \nabla J_{surr}(\pi_b).$$

Therefore, taking a gradient update with a proper step size on $J_{surr}$ can also improve the policy on the objective $J$. In our algorithm, the exponential weight poses an implicit constraint on the policy update similar to (Wang et al., 2018; Nair et al., 2020). With a proper constraint, optimizing on the $J_{surr}$ is equivalent as optimizing on the goal-conditioned objective $J$. Intuitively, $J_{surr}$ maximizes the likelihood for state-action pairs with large returns, and therefore it can lead to policy improvement based on the offline dataset.

### B.3  PROOF OF COROLLARY 1

**Corollary.** *Suppose function $h(s, a, g) \geq 1$ over the state-action-goal combination. Given trajectory $\tau = (s_1, a_1, \cdots, s_T, a_T)$, let $w_{t,i} = \gamma^{i-t} h(s_t, a_t, \phi(s_i)), t \in [1, T], i \in [t, T]$, then the following bounds hold:*

$$J_{surr}(\pi) \geq T \cdot J_{WGCSL}(\pi).$$

*Proof.* Similarly to the proof of Theorem 1, we have

$$J_{surr}(\pi) = \frac{1}{T} \mathbb{E}_{g \sim p(g), \tau \sim \pi_b(\cdot|g)} \left[ \sum_{t=1}^{T} \log \pi(a_t|s_t, g) \sum_{i=t}^{T} \gamma^{i-1} \cdot 1[\phi(s_i) = g] \right]$$

$$\geq \frac{1}{T} \mathbb{E}_{g \sim p(g), \tau \sim \pi_b(\cdot|g)} \left[ \sum_{t=1}^{T} \log \pi(a_t|s_t, g) \sum_{i=t}^{T} \gamma^{i-t} h(s_t, a_t, \phi(s_i)) \cdot 1[\phi(s_i) = g] \right]$$

$$\geq T \cdot \mathbb{E}_{g \sim p(g), \tau \sim \pi_b(\cdot|g), t \sim [1,T], i \sim [t,T]} \left[ \gamma^{i-t} h(s_t, a_t, \phi(s_i)) \log \pi(a_t|s_t, \phi(s_i)) \right]$$

$$\triangleq T \cdot J_{WGCSL}(\pi).$$

### B.4  PROOF OF PROPOSITION 1

**Proposition.** *Under certain conditions, $\tilde{\pi}$ is uniformly as good as or better than $\pi_{relabel}$. That is, $\forall s_t, \phi(s_i) \in D_{relabel}$ and $i \geq t$, $V^{\tilde{\pi}}(s_t, \phi(s_i)) \geq V^{\pi_{relabel}}(s_t, \phi(s_i))$, where $D_{relabel}$ contains the experiences after goal relabeling.*

*Proof.* We assume the advantage value is for the behavior policy of the relabeled data, i.e., $\pi_{relabel}$. We consider the compound state $\hat{s} = (s_t, \phi(s_i))$ from the relabeled distribution $D_{relabel}$. When $\hat{s}$ is fixed (the index of $i, t$ are fixed), $\gamma^{i-t}$ is a constant and we have:

$$f(\hat{s}, A(\hat{s}, a_t)) = clip(A(\hat{s}, a_t), -\infty, \log M) + \log(\gamma^{i-t}) + \log(\epsilon(A(\hat{s}, a_t))) + C(\hat{s})$$

$f(\hat{s}, \cdot)$ is a monotonically non-decreasing function for fixed $\hat{s}$, where $C(\hat{s})$ is a normalizing factor, $M$ is the positive maximum exponential advantage value, and $\epsilon(A(\hat{s}, a_t))$ is also a non-decreasing function introduced in Section 3.2 with minimum $\epsilon_{min} > 0$. Leveraging the Proposition 1 in MAR-WIL (Wang et al., 2018), we can conclude that $V^{\tilde{\pi}}(\hat{s}) \geq V^{\pi_{relabel}}(\hat{s})$.

**Remark.** The above proposition assumes the advantage value is for $\pi_{relabel}$. However, we empirically find that using advantage value of the learned policy $\pi$ (the value function is learned to minimize $\mathcal{L}_{TD}$ in Section 4) works better in offline goal-conditioned RL setting. As analyzed in prior works (Nair et al., 2020; Yang et al., 2021), using the value function of $\pi$ is equivalent to solving the following problem:

$$\pi_{k+1} = \arg\max_{\pi} \mathbb{E}_{g \sim p(g), a \sim \pi(\cdot|s,g)}[A^{\pi_k}(s, a, g)], \quad \text{s.t.} \quad D_{KL}(\pi(\cdot|s,g)\|\pi_{relabel}(\cdot|s,g)) \leq \epsilon.$$

In addition, for offline goal-conditioned RL, learning policies with inter-trajectory information is important for reaching diverse goals. Therefore, using $Q(s_{t+1}, \pi(s_{t+1}, g'), g')$ to compute TD target helps value estimation across trajectories compared with $Q(s_{t+1}, \pi_{relabel}(s_{t+1}, g'), g')$. Moreover, $\pi$ is trained via weighted supervised learning to avoid the OOD action problem.

## B.5   PROOF OF PROPOSITION 2

**Proposition.** *Assume the state space can be perfectly mapped to the goal space, $\forall s_t \in S, \phi(s_t) \in G$, and the dataset has sufficient coverage. We define a relabeling strategy for $(s_t, g)$ as*

$$g' = \begin{cases} \phi(s_i), i \sim U[t, T], & if\ \phi(s_i) \neq g, \forall i \geq t, \\ g, & otherwise. \end{cases}$$

*We also assume that when $g' \neq g$, the strategy only accepts relabeled trajectories with higher future returns than any trajectory with goal $g'$ in the original dataset $D$. Then, $\pi_{relabel}$ is uniformly as good as or better than $\pi_b$, i.e., $\forall s_t, g, \in D, V^{\pi_{relabel}}(s_t, g) \geq V^{\pi_b}(s_t, g)$, where $\pi_b$ is the behavior policy forming the dataset $D$.*

*Proof.* Let $D = \{\tau_j = (s_i, a_i, g_j)_{i=1}^T\}_{j=1}^N$. We assume that the data of $D$ and $D_{relabel}$ are sufficient and cover the full state and goal space, i.e., $\bigcup(s_i, g_j) = \mathcal{S} \times \mathcal{G}$. Note that every trajectory has only a single goal, and it does not have to reach the goal within the trajectory. Therefore, we can partition $D$ into two subsets $D^p$ (positive set) and $D^n$ (negative set):

$$D^p = \{\tau = (s_i, a_i, g_j)_{i=1}^T | \exists s_i \in \tau, \phi(s_i) = g_j\},$$
$$D^n = \{\tau = (s_i, a_i, g_j)_{i=1}^T | \forall s_i \in \tau, \phi(s_i) \neq g_j\},$$

where $D^p$ contains trajectories that can reach the goal and $D^n$ contains trajectories that cannot reach the goal. Let $D(s_t, g) = \{\tau = (s_i, a_i, g)_{i=t}^T | \tau \text{ begin with } s_t\}$, and similarly we define the partition

$$D^p(s_t, g) = \{\tau = (s_i, a_i, g)_{i=t}^T | \tau \text{ begin with } s_t, \exists s_i \in \tau, \phi(s_i) = g\},$$
$$D^n(s_t, g) = \{\tau = (s_i, a_i, g)_{i=t}^T | \tau \text{ begin with } s_t, \forall s_i \in \tau, \phi(s_i) \neq g\}.$$

Hence, the value $V^{\pi_b}$ can be written as

$$V^{\pi_b}(s_t, g) = \mathbb{E}_{\tau \sim D(s_t, g)}\left[\sum_{j=t}^T \gamma^{j-t} \cdot 1[\phi(s_j) = g]\right] \triangleq \mathbb{E}_{\tau \sim D(s_t, g)}[R_t^\tau]$$

$$= \frac{|D^p(s_t, g)|}{|D(s_t, g)|}\mathbb{E}_{\tau \sim D^p(s_t, g)}[R_t^\tau] + \frac{|D^n(s_t, g)|}{|D(s_t, g)|}\mathbb{E}_{\tau \sim D^n(s_t, g)}[R_t^\tau]$$

$$= \frac{|D^p(s_t, g)|}{|D(s_t, g)|}\mathbb{E}_{\tau \sim D^p(s_t, g)}[R_t^\tau].$$

Given $\tau = (s_i, a_i, g)_{i=t}^T \in D$, we relabel the future trajectory as $\tau_{relabel} = (s_i, a_i, g')_{i=t}^T$ for value estimation following the strategy

$$g' = \begin{cases} \phi(s_j), j \sim U[t, T], & \text{if } \forall j \geq t, \phi(s_j) \neq g \\ g, & \text{otherwise} \end{cases},$$

and when $g' \neq g$, the strategy only accepts $\tau_{relabel}$ with higher return than any trajectory in $D(s_t, g')$. Let $D_{relabel}(s_t, g) = \{\tau_{relabel} = (s_i, a_i, g)_{i=t}^{T} | \tau_{relabel}$ begin with $s_t\}$ be the relabeled trajectories with $g$ as goal. Hence, the value $V^{\pi_{relabel}}$ can be written as

$$V^{\pi_{relabel}}(s_t, g) = \mathbb{E}_{\tau_{relabel} \sim D_{relabel}(s_t, g)}[R_t^{\tau_{relabel}}].$$

For any $\tau_{relabel} \in D_{relabel}(s_t, g)$, we enumerate the situation of its counterpart $\tau$ without relabeling.

1. $\tau \in D^p(s_t, g)$. Following the relabeling strategy, we have $g' = g$, such that $\tau_{relabel} = \tau$.

2. $\tau \notin D^p(s_t, g)$, then $\tau$ has another goal $\tilde{g} \neq g$, but it reaches $g$ at some intermediate state. It holds that $R_t^{\tau_{relabel}} > 0 = R_t^{\tau}$. Following the relabeling strategy, we further have $R_t^{\tau_{relabel}} \geq R_t^{\tau'}, \forall \tau' \in D(s_t, g)$, which guarantees improvements on expected value of all trajectories starting from $s_t$ with goal $g$.

Therefore, we have

$$\begin{aligned} V^{\pi_{relabel}}(s_t, g) &= \mathbb{E}_{\tau_{relabel} \sim D_{relabel}(s_t, g)}[R_t^{\tau_{relabel}}] \\ &\geq \mathbb{E}_{\tau \sim D^p(s_t, g)}[R_t^{\tau}] \\ &\geq \frac{|D^p(s_t, g)|}{|D(s_t, g)|}\mathbb{E}_{\tau \sim D^p(s_t, g)}[R_t^{\tau}] = V^{\pi_b}(s_t, g). \end{aligned}$$

**Remark.** The relabeling strategy of Proposition 2 requires a complete scan of future states of trajectories, which is costly. Empirically we find that relabeling with random future states as HER has comparable performance, and is very simple and efficient to implement. We provide the comparison between the relabeling strategy of Proposition 2 (called WGCSL-slow-relabel) and the relabeling strategy that we used in practice in Table 2.

Table 2: Average Final Performance of Different Relabeling Strategies.

|  | WGCSL | WGCSL-slow-relabel |
| --- | --- | --- |
| PointRooms-expert | 36.15 ($\pm$0.85) | 37.08 ($\pm$0.56) |
| FetchReach-expert | 46.33 ($\pm$0.04) | 46.15 ($\pm$0.10) |
| FetchPush-expert | 39.11 ($\pm$0.17) | 39.26 ($\pm$0.28) |
| FetchSlide-expert | 10.73 ($\pm$1.09) | 9.34 ($\pm$1.71) |
| FetchPick-expert | 34.37 ($\pm$0.51) | 33.99 ($\pm$1.81) |
| HandReach-expert | 26.73 ($\pm$1.20) | 28.61 ($\pm$1.80) |
| PointRooms-random | 35.52 ($\pm$0.80) | 36.56 ($\pm$1.28) |
| FetchReach-random | 46.50 ($\pm$0.09) | 46.61 ($\pm$0.07) |
| FetchPush-random | 11.48 ($\pm$1.03) | 11.64 ($\pm$2.07) |
| FetchSlide-random | 2.34 ($\pm$0.28) | 1.81 ($\pm$0.41) |
| FetchPick-random | 8.06 ($\pm$0.71) | 6.48 ($\pm$2.24) |
| HandReach-random | 5.23 ($\pm$0.55) | 5.29 ($\pm$1.71) |

## C  ALGORITHM

We summarize our proposed approach WGCSL into Algorithm 1.

## D  OFFLINE SETTINGS

**Offline Datasets**  In the offline setting, we only use collected datasets to train agents, without additional interaction with the environment. The difference between the offline goal-conditioned RL dataset and the general offline RL dataset is that it also needs to save the original desired goals $g$ and the achieved goals $\phi(s_t)$. We follow previous works (Fujimoto et al., 2019; Fu et al., 2020) to introduce two different types of datasets:

---

**Algorithm 1:** Weighted Goal-Conditioned Supervised Learning

---

Given offline dataset $D$, the best advantage percentile threshold $N$;
Initialize policy $\pi_\theta$ and value function $Q$;
Initialize a First-In-First-Out queue $B = \{\}$ to save recent advantage values;
**for** *training step* $= 1, 2, \ldots, M$ **do**
   Sample a mini-batch from the offline dataset: $\{(s_t, a_t, g, r_t)\} \sim D$;
   Relabel the mini-batch using future goals:
    $\{(s_t, a_t, \phi(s_i), r(s_t, a_t, \phi(s_i))), i \geq t\} \sim D_{relabel}$;
   Update value function $Q$ to minimize TD error $\mathcal{L}_{TD}$ on the relabeled mini-batch;
   Estimate advantage value $A(s_t, a_t, \phi(s_i))$ using value function $Q$;
   Update queue $B$ with the estimated advantage values;

   Get the $N$ percentile advantage value $\hat{A}$ from $B$ to compute the BAW;
   Update policy $\pi_\theta$ to maximize the objective in Eq. 3 with the relabeled mini-batch:

$$J_{WGCSL} = \mathbb{E}_{(s_t, a_t, s_i) \sim D_{relabel}}[\gamma^{i-t} \log \pi_\theta(a_t | s_t, \phi(s_i)) exp_{clip}(A(s_t, a_t, \phi(s_i))) \cdot \epsilon(A(s_t, a_t, \phi(s_i)))]$$

**end for**

---

Table 3: Average Return of Different Offline Datasets

| Dataset | PointReach | PointRooms | Reacher | SawyerReach | SawyerDoor |
|---------|-----------|-----------|---------|-------------|------------|
| Random | 1.33 | 1.32 | 1.25 | 2.26 | 4.30 |
| Expert | 32.22 | 29.11 | 27.56 | 30.93 | 27.01 |
| **Dataset** | **FetchReach** | **FetchPush** | **FetchSlide** | **FetchPick** | **HandReach** |
| Random | 0.71 | 3.19 | 0.16 | 1.76 | 0.00 |
| Expert | 36.69 | 31.35 | 1.58 | 17.44 | 0.50 |

- Random dataset: using a uniformly random policy to collect trajectories.

- Expert dataset: using the final online policy trained by HER with additional Gaussian noise (zero mean and 0.2 standard deviation) and random actions (with probability 0.3) following previous work (Plappert et al., 2018) for data collection.

For the dataset of six easy tasks (PointReach, PointRooms, Reacher, SawyerReach, SawyerDoor, FetchReach), we save a dict of $1 \times 10^5$ transitions (2000 trajectories) containing observations, actions, desired goals and achieved goals. Regarding the other four hard tasks (FetchPush, FetchSlide, FetchPick, HandReach), we store $2 \times 10^6$ transitions (40000 trajectories) in the offline dataset. The rewards can be computed by leveraging the corresponding interface (namely *env.compute_reward*) of the environment. The average returns of the two types of datasets are shown in Table 3. Generally, average returns of the expert datasets are close to that of the optimal policy, while average returns of the random datasets are close to 0.

**Baseline Introduction**    In the offline goal-conditioned setting, we denote the offline dataset as $D$ and consider the following baselines:

- **GCSL**: GCSL (Ghosh et al., 2021) without online interaction, only using offline samples after relabeling (denoted as $D_{relabel}$) to maximize

$$J_{GCSL}(\pi) = \mathbb{E}_{(s_t, a_t, \phi(s_i)) \sim D_{relabel}}[\log \pi(a_t | s_t, \phi(s_i))].$$

- **Goal MARWIL**: goal-conditioned MARWIL (Wang et al., 2018), using offline samples to maximize

$$J_{MARWIL} = \mathbb{E}_{(s_t, a_t, g) \sim D}[\log \pi(a_t | s_t, g) \exp(\frac{1}{\beta} A(s_t, a_t, g))].$$

To fairly compare with our method, Goal MARWIL is implemented with actor-critic style as AWAC (Nair et al., 2020) and WGCSL, therefore it can also be treated as 'Goal AWAC'. Besides, we also clip the exponential advantage value as WGCSL for numerical stability.

- **Goal AWR**: goal-conditioned AWR (Peng et al., 2019), using offline samples to maximize

$$J_{AWR}(\pi) = \mathbb{E}_{(s_t,a_t,g)\sim D}[\log \pi(a_t|s_t,g)\exp(\frac{1}{\beta}(R_t - V(s_t,g)))],$$

where $R_t = \sum_{i=0}^{T-t} \gamma^i r_{t+i}$. Different from the original AWR that uses TD($\lambda$) or Monte Carlo return $R_t$ to train the value function, we use TD loss to learn value function as WGCSL. Empirically, we find TD target works better than Monte Carlo in our setting.

- **Goal BC**: goal-conditioned behavior cloning (Bain & Sammut, 1995), using offline samples to maximize

$$J_{BC}(\pi) = \mathbb{E}_{(s_t,a_t,g)\sim D}[\log \pi(a_t|s_t,g)].$$

- **Goal BCQ**: goal-conditioned Batch-Constrained Q-Learning (Fujimoto et al., 2019), which we use the official codebase and concatenate observations, desired goals and achieved goals as states. The objective function of Goal BCQ is as follows:

$$J_{BCQ}(\xi) = \mathbb{E}_{(s_t,g)\sim D,a\sim G_w(s_t,g)}[Q(s_t, a + \xi(s_t,a,g,\Phi),g)],$$

where $\xi(s_t,a,g,\Phi)$ is the goal-conditioned perturbation model, which outputs an adjustment to an action $a$ within the range $[-\Phi, \Phi]$. $G_w$ is the VAE fitted on the behavior policy of the offline dataset. The policy is optimized through optimizing $\xi$. The $Q$ function is learned using the Clipped Double Q-learning in (Fujimoto et al., 2019), and the only difference is that the $Q$ function also includes goal $g$ as input.

- **Goal CQL**: goal-conditioned Conservative Q-Learning (Kumar et al., 2020), which we also use the official codebase and concatenate observations, desired goals and achieved goals as states. The objective function of Goal CQL is as follows:

$$J_{CQL}(\pi) = \mathbb{E}_{(s_t,g)\sim D,a\sim\pi(\cdot|s_t,g)}[Q(s_t,a,g) - \log \pi(a|s_t,g)],$$

which jointly optimizes estimated return and policy entropy. The Q function of CQL is learned by minimizing the following loss:

$$L_{CQL} = \alpha\mathbb{E}_{(s_t,g)\sim D}\big[\log\sum_a \exp(Q(s_t,a,g)) - \mathbb{E}_{a\sim\pi_b(\cdot|s_t,g)}[Q(s_t,a,g)]\big]$$
$$+ \frac{1}{2}\mathbb{E}_{(s_t,a_t,s_{t+1},g)\sim D}\big[(Q(s_t,a_t,g) - \mathcal{B}^\pi Q(s_t,a_t,g))^2\big],$$

where $\pi_b$ is the behavior policy for the offline dataset and $\mathcal{B}^\pi$ is the Bellman operator.

- **HER**: Hindsight Experience Replay (Andrychowicz et al., 2017), using offline samples after relabeling to maximize

$$J_{HER}(\pi) = \mathbb{E}_{(s_t,a_t,g')\sim D_{relabel}}[Q(s_t,\pi(s_t,g'),g')],$$

where $g' = \phi(s_i), i \geq t$ and the $Q$ function is learned by minimizing TD error:

$$L_{TD} = \mathbb{E}_{(s_t,a_t,s_{t+1},g')\sim D_{relabel}}(r_t + \gamma Q(s_{t+1},\pi(s_{t+1},g'),g') - Q(s_t,a_t,g'))^2.$$

**Implementation of Baselines** For fair comparison, all the implementation of WGCSL, GCSL, Goal MARWIL, Goal AWR, Goal BC, and HER use the Diagnal Gaussian policy with a mean vector $\pi_\theta(s,g)$ and a non-zero constant variance $\sigma^2$, where $\sigma$ is set as 0.2. WGCSL, Goal MARWIL, Goal AWR, and HER share the same network structure, i.e., the value function and the policy network (along with their target networks) are both 3-layer MLPs with 256 units each layer and relu activation. Besides, GCSL and Goal BC both keep a single policy network without target network and value function. We also normalize the observations and goals with estimated mean and standard deviation for WGCSL and other baselines, which is helpful for offline multi-goal manipulation tasks. The parameter $\beta$ in Goal MARWIL and Goal AWR are set as 1 by default. As regards to Goal BCQ and Goal CQL, we use the same batch size as other algorithms and other parameters are set as default (we find that default parameters perform equally well, if not better than others, in the grid search of critical parameters). All other hyper-parameters of different algorithms are kept the same for each task, such as batch size (128 for the first 6 tasks and 512 for 4 harder taks), discount factor $\gamma = 0.98$, and Adam optimizer with learning rate $5 \times 10^{-4}$.

**Implementation of WGCSL** The basic implementation is introduced as above. To compute the best-advantage weight for WGCSL, we keep a First-In-First-Out queue $B$ of size $5 \times 10^4$ to store recent calculated advantage values, and the percentile threshold $N$ gradually increases from 0 to 80. For each training step, $N$ increases by 0.01 (for 4 harder tasks from FetchPush to HandReach) or 0.15 (for other tasks). The maximum clipping upper bound $M$ for WGCSL is set as 10. Though DRW brings improvements for most of the tasks on expert and random datasets, we find DRW slightly affects the performance of harder tasks on random datasets, which may be because in the random datasets we need more timesteps to obtain enough coverage of achieved goals, especially for harder tasks. Tuning $\gamma$ for DRW trade-offs between the value and the effective coverage of relabeling goals, and excluding DRW is equivalent to setting $\gamma = 1$ for DRW. Therefore, we only use GEAW and BAW for these tasks on random datasets.

**Evaluation Settings** For evaluation, we evaluate all the algorithms for 100 episodes without randomness. We repeat for 5 different random seeds and report the average evaluation performance with standard deviation.

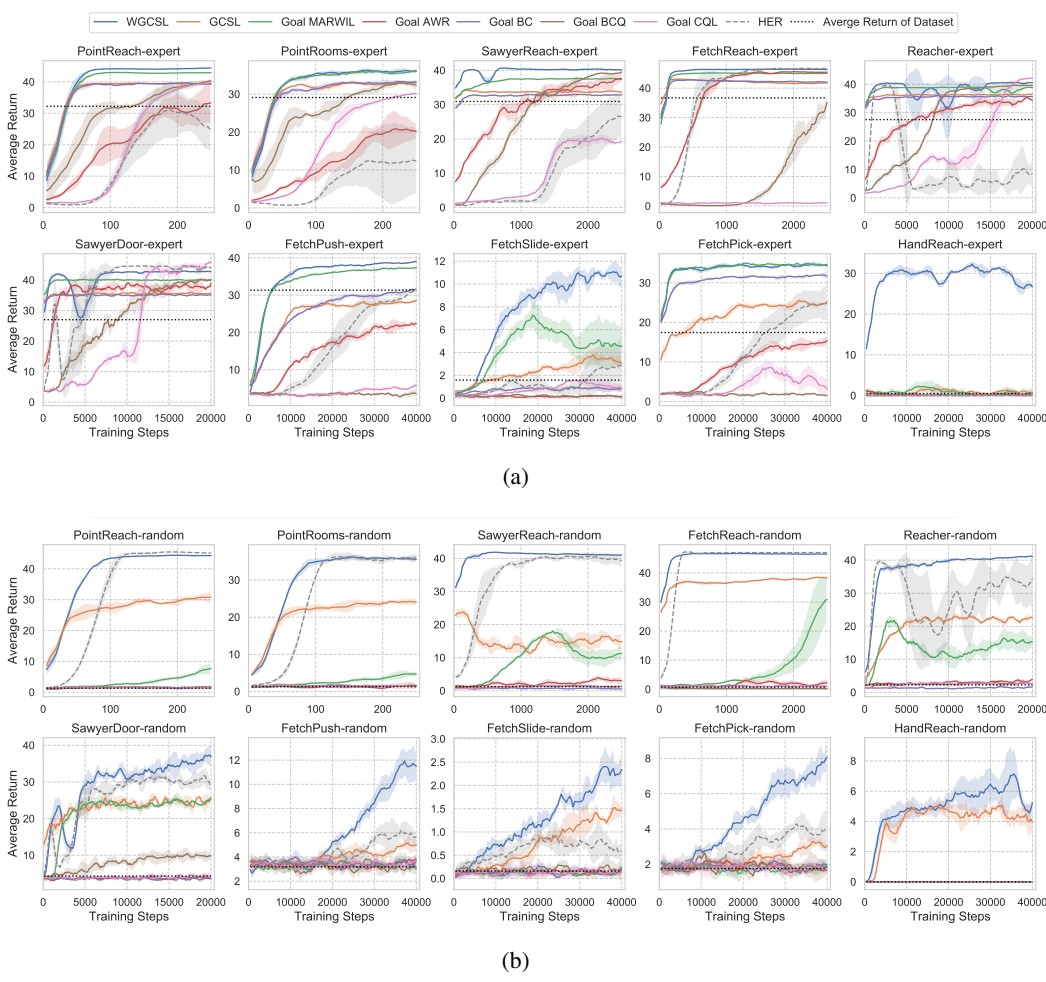

Figure 7: Full average returns of different algorithms on (a) expert and (b) random datasets.

# E ADDITIONAL EXPERIMENTAL RESULTS

In this section, we provide more empirical results in both offline and online goal-conditioned RL setting, including average return, final distance of the offline setting, results on small (10%) dataset, and the results of online experiments.

## E.1 FULL OFFLINE RESULTS

We provide the average return curves of all tasks in Figure 7. In these results, WGCSL performs consistently well across ten environments, especially when handling hard tasks (FetchPush, FetchSlide, FetchPick, HandReach) and learning from the random offline dataset. Other conclusions have been discussed in Section 5.2. Comparing AWR with MARWIL, the major difference is using Monte Carlo return to replace the TD estimation for learning the policy. Hence, we can find that TD bootstrap learns more efficiently in weighted supervised learning, which is similar to the conclusion of prior work (Nair et al., 2020). In addition, we present all final performance of different algorithms in Table E.9.

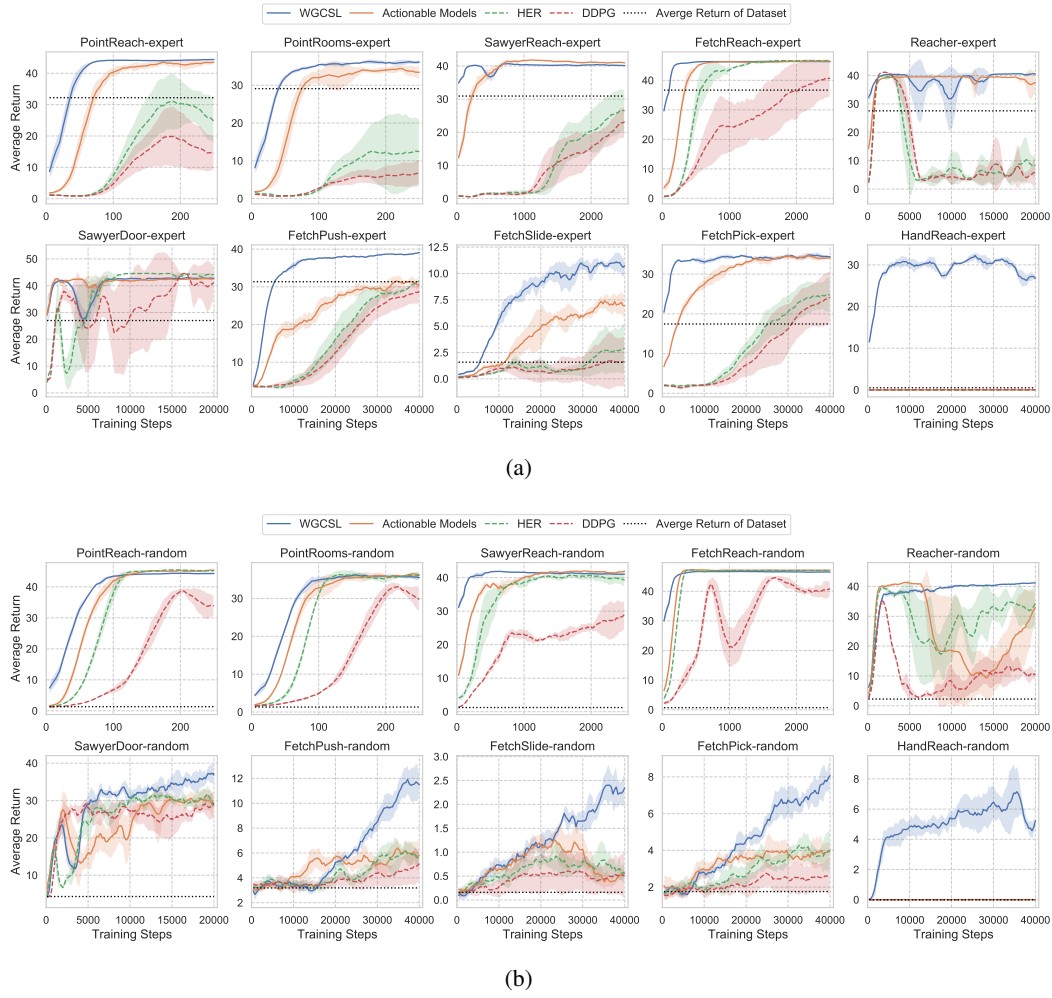

Figure 8: Comparison of WGCSL, Actionable Models, HER, DDPG on offline (a) expert and (b) random datasets.

## E.2 COMPARISON RESULTS WITH ACTIONABLE MODELS, HER AND DDPG

In this subsection, we make comparison with Actionable Models (AM) (Chebotar et al., 2021), which applies the conservative estimation idea similar to (Kumar et al., 2020) into offline goal-conditioned RL. There are some different settings between AM and ours. For example, we consider cumulative return of the maximum horizon, i.e., the agent doesn't terminate the interaction after reaching the goal. To fairly compare with AM, we implement AM based on HER. Specifically, we

revise the value update as:

$$\mathcal{L}_{AM} = \mathbb{E}_{(s_t,a_t,g',r'_t,s_{t+1})\sim D_{relabel}}[(r'_t + \gamma\hat{Q}(s_{t+1},\pi(s_{t+1},g'),g')) - Q(s_t,a_t,g'))^2]$$
$$+ \mathbb{E}_{(s_t,g')\sim D_{relabel},a\sim exp(Q)}[Q(s_t,a,g')]$$

In the formula, AM tries to minimize TD error and minimize unseen Q values at the same time. The action $a$ is sampled according to exponential Q values. In our implementation, we sample 20 actions from the action space and compute their exponential Q value in order to sample $a$.

The results are shown in Figure 8. We can observe that DDPG and HER are unstable during training, with large variance and performance drops at the end. In both expert and random dataset, HER outperforms DDPG, which demonstrates the benefits of goal relabeling for offline goal-conditioned RL. Besides, Actionable Model (AM) achieves higher final performance than HER and DDPG especially in expert dataset, which indicates the conservative estimation technique is useful for offline goal-conditioned RL. Moreover, WGCSL is more efficient than AM, especially in harder tasks with large action space. Intuitively, WGCSL promotes the probability of visited actions, while AM restrains the probability of unvisited actions. For tasks with large action space, conservative strategy can be ineffective compared to WGCSL.

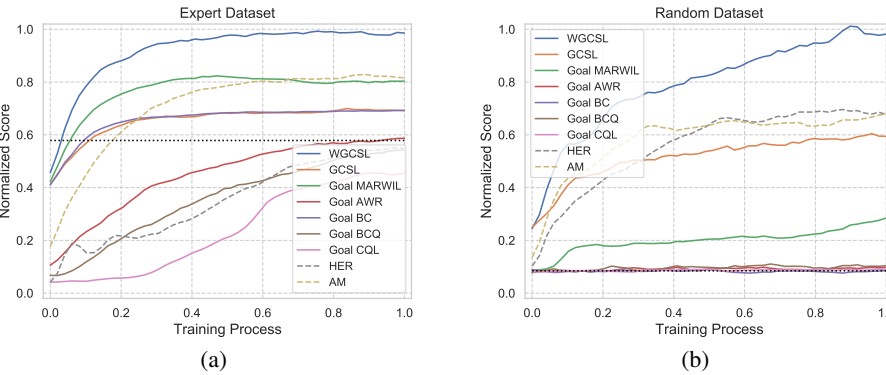

Figure 9: Normalized score across 10 tasks on (a) expert and (b) random datasets. The black dotted line refers to the normalized score of offline dataset. The value are normalized by dividing by the maximum final return of each task in Table 1. Note that the value can exceed 1 as the final return can be lower than the return during training. The x-axis represents the training process scaled to 1 for different tasks.

### E.3 Normalized Score Across 10 Tasks

For clarity, we report the normalized score across 10 tasks to compare different algorithms in Figure 9. We can observe the significant improvement of WGCSL over other baselines. We take an insight into the impact of the goal relabeling technique. For expert datasets, goal relabeling is not a dominant factor, e.g., the learning curve of Goal BC and GCSL are very close. In random datasets, goal relabeling is essential to achieve a high performance. In addition to goal relabeling, WGCSL contains other techniques to jointly promote the performance in offline goal-conditioned RL tasks. Other conclusions are the same as prior subsections (Appendix E.1 and Appendix E.2).

### E.4 Comparison of Training Time

We further analyze the computation cost of the four most effective algorithms in the random offline setting, i.e., WGCSL, GCSL, HER and Actionable Models (AM). The results are shown in Figure 10, from which we can conclude that GCSL needs the minimal training time, while WGCSL and HER require similar cost. It needs to be emphasized that WGCSL has only a small amount of additional training time over GCSL, but it achieves significantly higher performance. Therefore, we believe that the cost of learning the value function on top of GCSL is worthwhile. Moreover, AM requires the largest computation among these algorithms, which is mainly because it needs to

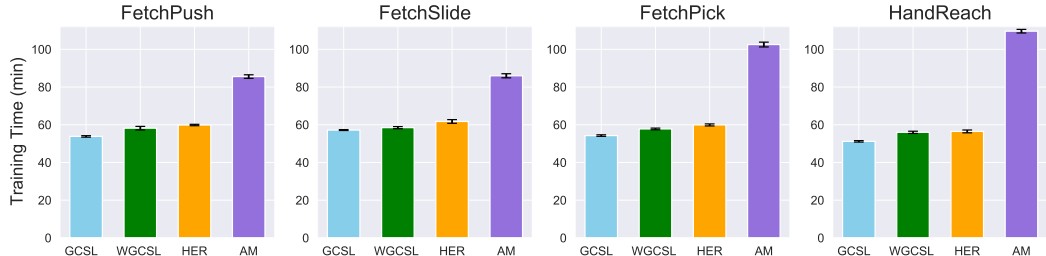

Figure 10: Training time of GCSL, WGCSL, HER and Actionable Models (AM) on 4 harder tasks.

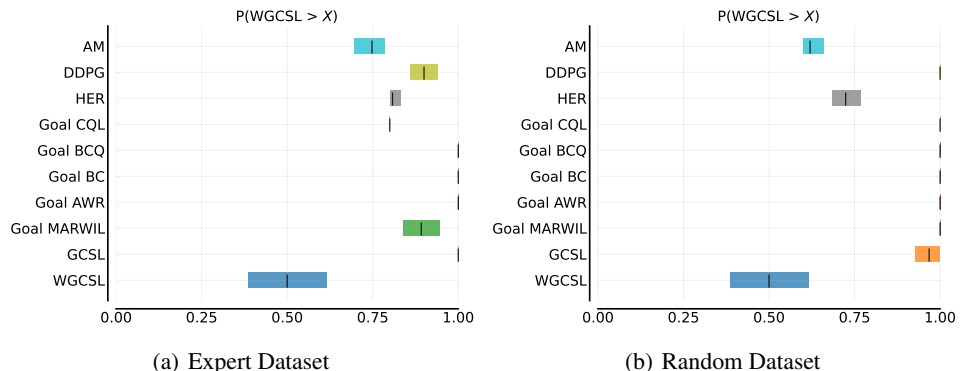

(a) Expert Dataset                    (b) Random Dataset

Figure 11: Average probability of improvement on offline (a) expert and (b) random datasets. Each figure shows the probability of improvement of WGCSL compared to all other algorithms. The interval estimates are based on stratified bootstrap with independent sampling with 2000 bootstrap re-samples.

sample several actions and compute their exponential Q value for constraining the large Q value. These procedures make AM the most time-consuming algorithm. In contrast, WGCSL can achieve higher performance with less training time. The training time would vary across different platforms, and we use one single GPU (Tesla P100 PCIe 16GB) and 5 cpu cores (Intel Xeon E5-2680 v4 @ 2.40GHz) to run 5 random seeds parallelly.

### E.5 AVERAGE PROBABILITY OF IMPROVEMENT

In this subsection, we adopt the average probability of improvement (Agarwal et al., 2021), a robust metric to measure how likely it is for one algorithm to outperform another on a randomly selected task. The results are reported in Figure 11. As shown in the results, WGCSL robustly outperforms other baselines in both expert and random datasets. For instance, WGCSL is $100\%$ better than 4 baselines on the expert dataset, and 6 baselines on the random dataset. Note that the most two effective and robust algorithms on both random and expert datasets are WGCSL and Actionable Models (AM), which are specifically designed for offline goal-conditioned RL. Comparing the two algorithms, WGCSL outperforms AM with a probability of $75\%$ on the expert dataset and $62\%$ on the random dataset.

### E.6 FINAL DISTANCES IN THE OFFLINE SETTING

GCSL (Ghosh et al., 2021) uses the average final distance to the desired goal as a measure for goal-reaching problems. We also provide average final distances in the offline setting to evaluate the learned goal-conditioned policy in Figure 12. From the results we can conclude that WGCSL substantially outperforms other baselines in both random and expert dataset. Although GCSL learns a suboptimal policy, it also achieves relatively consistent final distance across different environments

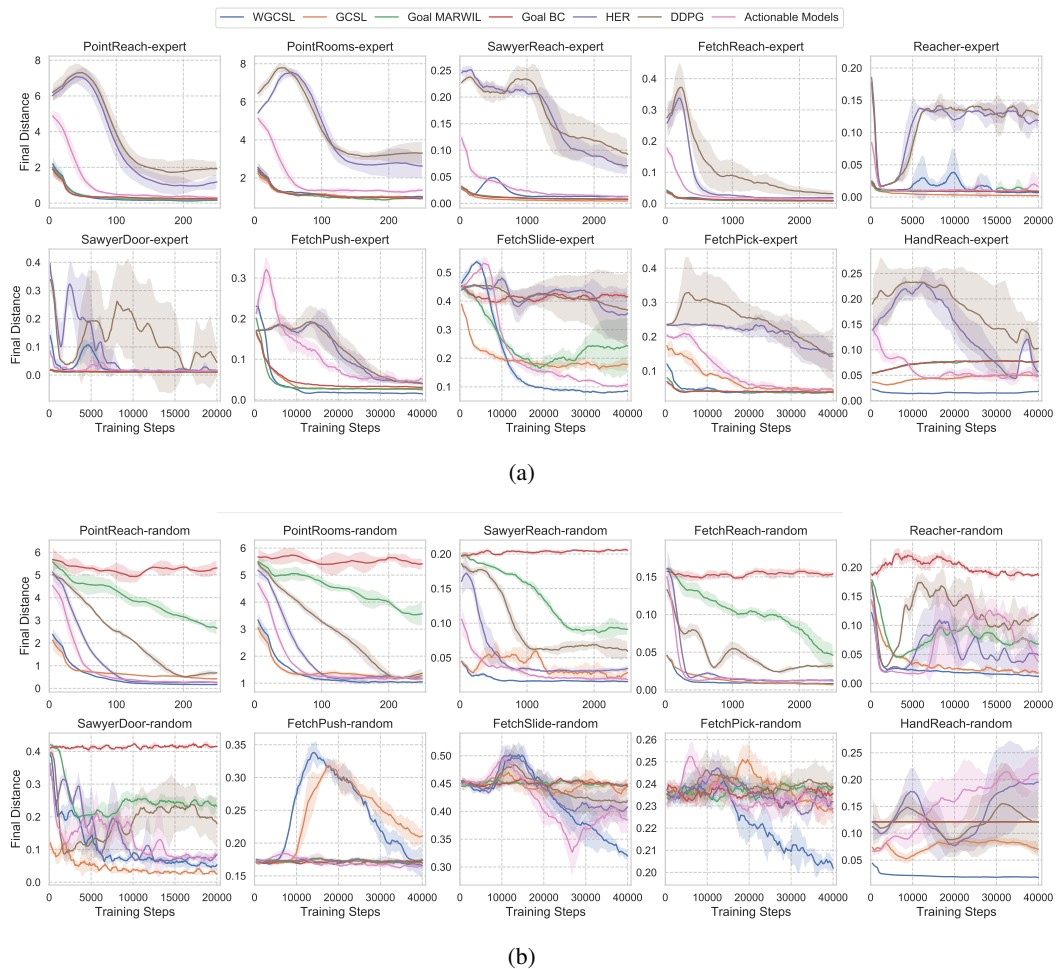

Figure 12: Average final distance of different algorithms on offline (a) expert and (b) random datasets.

and offline datasets. Besides, Goal MARWIL and Goal BC are two effective baselines when learning from the expert dataset, but they do not perform well on the random dataset. In addition, HER and DDPG learn unstable results mainly because of bootstrapping from out-of-distribution actions.

### E.7 RESULTS ON 10% OFFLINE DATASET

To evaluate the impact of dataset size on the performance, we select 10% of samples (i.e., $2 \times 10^5$ transitions) from the offline dataset of 4 hard tasks (FetchPush, FetchSlide, FetchPick, HandReach). The results are reported in Figure 13 and Figure 14. From the results, we can conclude that with only 10% offline data WGCSL consistently achieves the best performance out of these algorithms. In Figure 14, we can see that WGCSL is very robust to the dataset size on the expert dataset, while the performance is affected more by the small data size on the random dataset. The probable reason is that in the offline setting, the number of samples influences the data coverage greatly and learning generalizable value function from a small set of data is very difficult.

### E.8 ONLINE EXPERIMENTS

We introduce implementation details and hyper-parameters of the online setting in this subsection. As GCSL is originally proposed for online goal-conditioned RL, WGCSL can also be applied to online goal-conditioned setting. In the online training setting, the agent iterates between collecting trajectories from the environment and training with samples from the replay buffer. We conduct

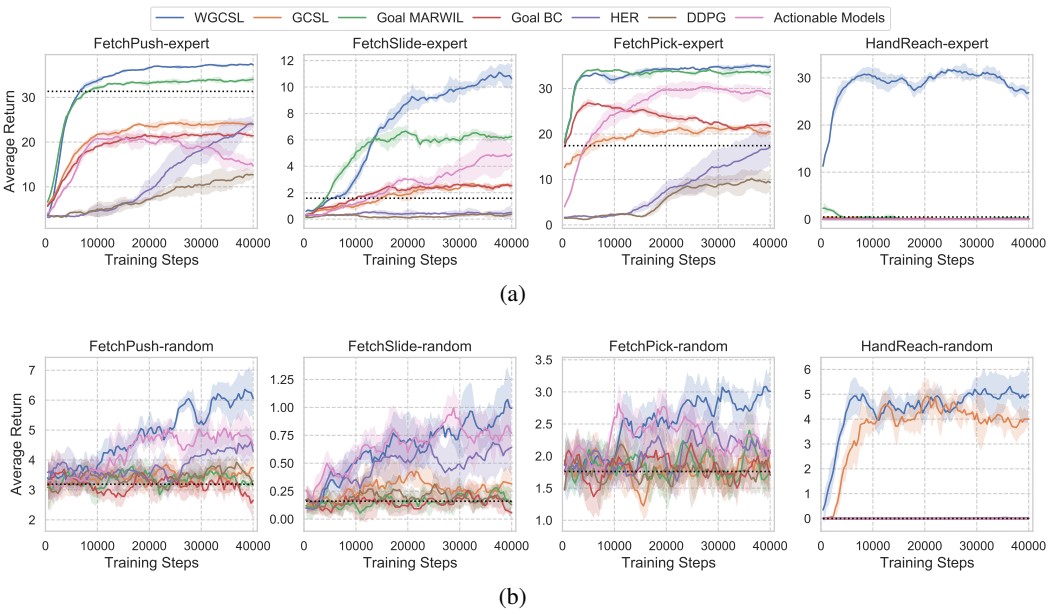

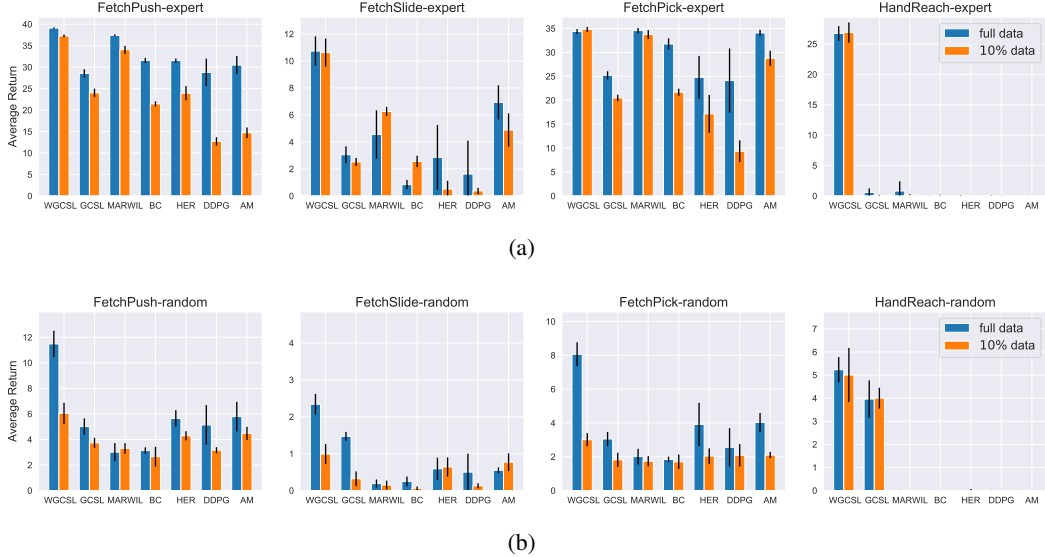

Figure 13: Average return of $10\%$ offline (a) expert and (b) random dataset.

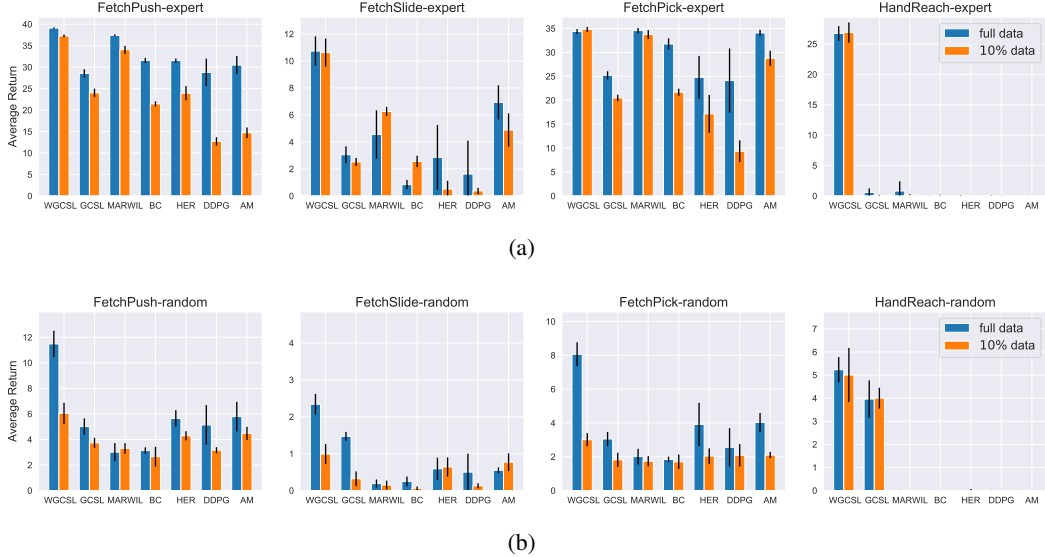

Figure 14: Final performance comparison of full dataset and $10\%$ dataset. Top row is the result on expert dataset, and the bottom row is the result on random dataset.

experiments on the first six taks, i.e., PointReach (here we call Point2DLargeEnv), PointRooms (here we call Point2D-FourRooms), SawyerReach, FetchReach, Reacher and SawyerDoor. The buffer size for first four tasks is set as $1 \times 10^4$, and $5 \times 10^4$ for the other two tasks. Before training starts, we first randomly sample 20 trajectories to initialize the buffer. The policies of WGCSL and GCSL are both Diagonal Gaussian policy with a mean vector $\pi_\theta(s, g)$ parameterized as $\theta$ and a non-zero constant variance $\sigma^2$. The standard deviation $\sigma$ is set as 0.2. All the policy networks and value networks are the same as that used in the offline setting. When interacting with the environments, we use additional random actions for exploration with a probability of 0.3. After data collection, we sample a mini-batch from the replay buffer and perform goal relabeling for WGCSL and GCSL. Then we train the agent for 1 mini-batch (Point2DLarge, Point2D-FourRooms) or 4 mini-batches

for other tasks. The batch size, optimizer, learning rate keeps the same as the offline setting. The maximum clipping upper bound $M$ for WGCSL is set as 10. We also utilize target nets for WGCSL to stabilize training, and the polyak average coefficient for target nets are 0.9. With respect to the evaluation phase, we use the deterministic policy $\pi_\theta(s_t, g)$ without randomness. After each epoch, we evaluate the policy for 100 episodes and compute the average success rate. For all the experiments we repeat for 10 different random seeds and report the mean performance with standard deviation. In this subsection, we report the results of WGCSL with the discounted relabeling weight (DRW) and the goal-conditioned exponential advantage weight (GEAW), though the performance can be improved with the best-advantage weight (BAW).

The empirical results regarding the average success rate of the online setting are shown in Figure 15. From Figure 15 we can conclude that WGCSL achieves better success rate and converges faster than other baselines in most of the environments except SawyerReach. Therefore, WGCSL is verified to be highly efficient in both online and offline goal-conditioend RL, and we hope it could be a stepping stone for future real-world RL applications.

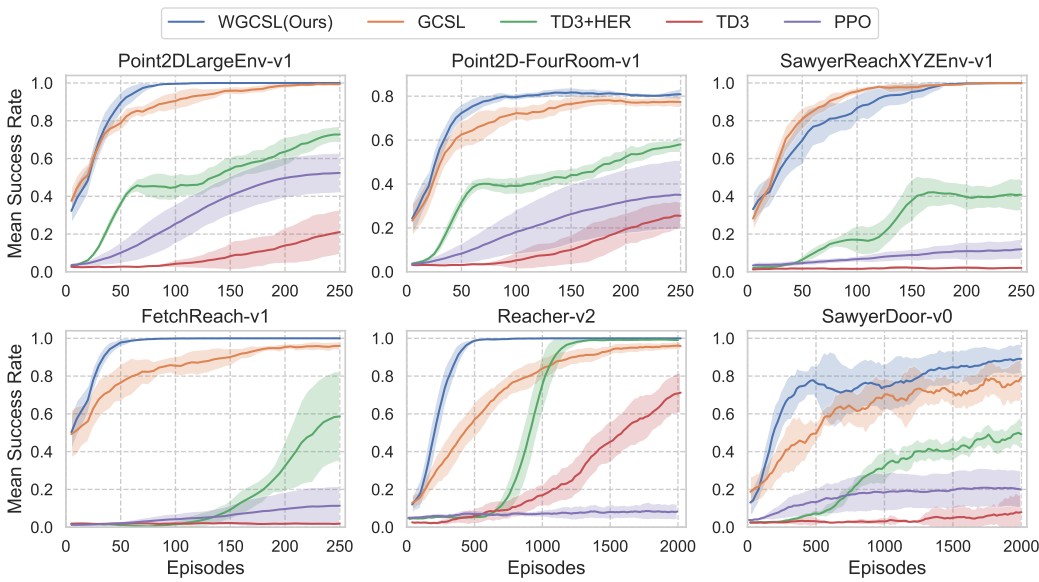

Figure 15: Mean success rate of online experiments with standard deviation (shaded) across 10 random seeds.

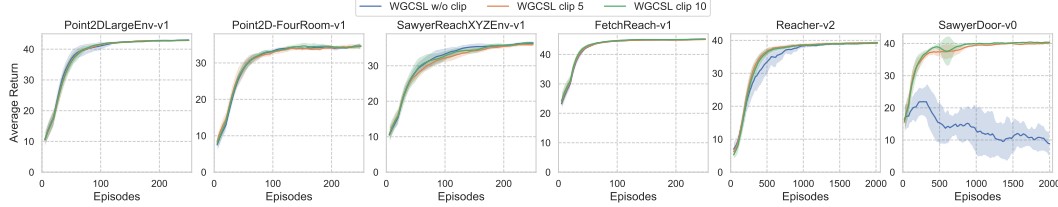

Figure 16: Study of the impact of clipping exponential advantage in WGCSL in the online setting. Results are averaged across 10 random seeds.

### E.9 THE IMPACT OF CLIPPING FOR EXPONENTIAL ADVANTAGE WEIGHTS

We study the impact of clipping in WGCSL's exponential advantage weights by comparing with the following variants:

- WGCSL w/o clip: no clipping weights performed.

- WGCSL clip 5: clipping the exponential advantage weights to $(0, 5]$.

- WGCSL clip 10: clipping the exponential advantage weights to $(0, 10]$, which is used in our paper by default.

From the results in Figure 16 we can observe that without clipping the advantage estimation can harm the performance in Reacher and SawyerDoor. Therefore, we perform weight clipping by default.

## F    DETAILS OF ENVIRONMENTS

All of the ten tasks have continuous state space, action space and goal space. The maximum episode horizon is set as 50. In this section, we introduce these tasks in detail.

**PointReach**    PointReach is taken from the open-source environment *multi-world*[2] and it requires the blue point to reach the green circle. The state space $[-5, 5] \times [-5, 5]$ has two dimensions representing Cartesian coordinates of the blue point, and the action space $[-1, 1] \times [-1, 1]$ also has two dimensions meaning the horizontal and vertical displacement. The goal space is the same as state space, which means $\phi(s) = s$. The bule point and the green circle are randomly initialized in the state space. The allowable error $\epsilon$ of reaching goals is the radius of the target circle and is set as 1. The reward function is defined as:

$$r(s_{XY}, a, g_{XY}) = 1(\|s_{XY} - g_{XY}\|_2^2 \le \epsilon).$$

**PointRooms**    The PointRooms environment is built on *multi-world*. The state space, the action space, the goal space, and the reward function are the same as PointReach. The difference is that there are four rooms separated by walls.

**Reacher**    The Reacher environment is revised from OpenAI Gym (Brockman et al., 2016). States are 11-dimensional, which indicate the angles, the positions, and the velocity of the joints. Actions are 2-dimensional and they control the movement of two joints. Goals are 2-dimensional vectors representing the expected XY position. And the state-to-goal mapping is $\phi(s) = s[-3 : -1]$, where the last three dimensions are the XYZ position of the end-effect. The reward function is defined as:

$$r(s_{XY}, a, g_{XY}) = 1(\|s_{XY} - g_{XY}\|_2^2 \le \epsilon),$$

where the allowable error $\epsilon$ is set as 0.05.

**SawyerReach**    The SawyerReach environment is taken from *multi-world*. The Sawyer robot aims to reach a target position with its end-effector. The observation space is 3-dimensional, representing the 3D Cartesian position of the end-effector. Correspondingly, the goal space is 3-dimensional and describes the expected position, and the state-to-goal mapping is $\phi(s) = s$. Besides, the action space has 3 dimensions describing the next position of the end-effector. The reward function is defined as:

$$r(s_{XYZ}, a, g_{XYZ}) = 1(\|s_{XYZ} - g_{XYZ}\|_2^2 \le \epsilon),$$

where the allowable error $\epsilon = 0.06$.

**SawyerDoor**    The SawyerDoor environment is revised from *multi-world*. The Sawyer robot is required to open the door to a desired angle. The state space (4-dimensional) consists of the Cartesian coordinates of the Sawyer end-effector and the door's angle. As in the SawyerReach task, the action space is 3-dimensional and controls the position of the end-effector. The desired goals are uniform from 0 to 0.83 radians. And the reward function is defined as:

$$r(s, a, g_{angle}) = 1(|\phi(s) - g_{angle}| \le \epsilon),$$

where the allowable error $\epsilon$ is set as 0.06.

---

[2]https://github.com/vitchyr/multiworld

**FetchReach**    The FetchReach environment is taken from OpenAI Gym (Brockman et al., 2016; Plappert et al., 2018). In this environment, a 7-DoF robotic arm is expected to touch a desired location with its two-finger gripper. The state space is 10-dimensional, including the gripper's position and linear velocities. The action space is 4-dimensional, which represents the gripper's movements and its status about the opening and closing. Moreover, the goals are 3-dimensional vectors representing the target places of the gripper. The state-to-goal mapping is $\phi(s) = s[0:3]$, because the first 3 dimensions of the state describe the position of the gripper. The allowable error in FetchReach is $\epsilon = 0.05$. The reward function is defined as:

$$r(s, a, g_{XYZ}) = 1(\|\phi(s) - g_{XYZ}\|_2^2 \leq \epsilon).$$

**FetchPush**    Similar to FetchReach, the FetchPush environment is taken from OpenAI Gym where a 7-DoF robotic arm is controlled to move a box to the desired location. The state space is 25-dimensional, including the gripper's position, linear velocities, and the box's position, rotation, linear and angular velocities. The action space is also 4-dimensional as in FetchReach. Different from FetchReach, the 3-dimensional goals and achieved goals represent the target and current position of the box. The state-to-goal mapping is $\phi(s) = s[3:6]$, because the 3 dimensions of state describe the position of the box. The allowable error in FetchPush is $\epsilon = 0.05$. The reward function is defined as:

$$r(s, a, g_{XYZ}) = 1(\|\phi(s) - g_{XYZ}\|_2^2 \leq \epsilon).$$

**FetchSlide**    The FetchSlide environment is like FetchPush where a 7-DoF robotic arm is controlled to slide a box to the desired location. The state space, the action space, the state-to-goal mapping, the reward function and the allowable error are all the same as FetchPush. The difference is that the target position is outside of the robot's reach thus it has to hit the box and makes the box slide and then stop at the desired position.

**FetchPick**    Like FetchPush and FetchSlide environments, the FetchPick environment controls a 7-DoF robotic arm to pick up a box to the desired location. The state space, the action space, the state-to-goal mapping, the reward function and the allowable error are all the same as FetchPush. The difference is that the target position may be on the table or in the air, thus the agent needs to pick the box with its gripper.

**HandReach**    In HandReach environment, the agent is required to control a 24-DoF anthropomorphic hand and use its fingers to reach the target place. Observations is 63-dimensional containing two 24-dimensional vectors about positions and velocities of the joints, and additional 15 dimensions indicating current state of fingertips. Actions are 20-dimensional vectors that control the non-coupled joints of the hand. Goals have 15 dimensions for HandReach representing the target Cartesian positions of each fingertip. The reward function is the same as Fetch tasks except the state-to-goal-mapping ($\phi = s[-15:]$) and the allowable threshold ($\epsilon = 0.01$).

Table 4: Average final performance of all algorithms with standard deviation over 5 random seeds.

| Task Name | WGCSL | GCSL | g-MARWIL | g-AWR | g-BC | g-BCQ | g-CQL | DDPG | HER | AM |
|---|---|---|---|---|---|---|---|---|---|---|
| PointReach-expert | **44.40** (±0.14) | 39.27 (±0.48) | **42.95** (±0.15) | 33.35 (±6.64) | 39.36 (±0.48) | 40.42 (±0.57) | 39.75 (±0.33) | 14.53 (±5.36) | 24.68 (±7.07) | **43.56** (±0.69) |
| PointRooms-expert | **36.15** (±0.85) | 33.05 (±0.54) | **36.02** (±0.57) | 20.24 (±2.15) | 33.17 (±0.52) | 32.37 (±1.77) | 30.05 (±0.38) | 6.83 (±3.63) | 12.41 (±8.59) | 33.45 (±1.91) |
| Reacher-expert | **40.57** (±0.20) | 36.42 (±0.30) | 38.89 (±0.17) | 34.30 (±0.65) | 35.72 (±0.37) | 39.57 (±0.08) | **42.23** (±0.12) | 6.16 (±6.40) | 8.27 (±4.33) | 37.48 (±4.15) |
| SawyerReach-expert | **40.12** (±0.29) | 33.65 (±0.38) | 37.42 (±0.31) | 37.54 (±1.62) | 32.91 (±0.31) | **39.49** (±0.45) | 19.33 (±0.45) | 23.18 (±4.83) | 26.48 (±6.23) | **40.91** (±0.29) |
| SawyerDoor-expert | 42.81 (±0.23) | 35.67 (±0.09) | 40.03 (±0.16) | 38.91 (±0.63) | 35.03 (±0.20) | 40.13 (±0.75) | 41.03 (±0.11) | 41.03 (±6.74) | **44.09** (±0.65) | 42.49 (±0.54) |
| FetchReach-expert | **46.33** (±0.04) | 41.72 (±0.31) | **45.01** (±0.11) | **45.37** (±0.24) | 42.03 (±0.25) | 35.18 (±3.09) | **45.86** (±0.11) | 40.70 (±6.38) | **46.73** (±0.14) | **46.50** (±0.09) |
| FetchPush-expert | **39.11** (±0.17) | 28.56 (±0.96) | **37.42** (±0.22) | 22.44 (±1.70) | 31.56 (±0.61) | 3.62 (±0.96) | 5.76 (±0.83) | 28.76 (±3.21) | 31.53 (±0.47) | 30.49 (±2.11) |
| FetchSlide-expert | **10.73** (±1.09) | 3.05 (±0.62) | 4.55 (±1.79) | 0.14 (±0.13) | 0.84 (±0.35) | 0.12 (±0.10) | 0.86 (±0.38) | 1.62 (±2.47) | 2.86 (±2.40) | 6.92 (±1.26) |
| FetchPick-expert | 34.37 (±0.51) | 25.22 (±0.85) | **34.56** (±0.54) | 15.37 (±0.99) | 31.75 (±1.19) | 1.46 (±0.29) | 3.23 (±2.52) | 24.12 (±6.70) | 24.79 (±4.49) | **34.07** (±0.65) |
| HandReach-expert | **26.73** (±1.20) | 0.57 (±0.68) | 0.81 (±1.59) | 0.12 (±0.04) | 0.06 (±0.03) | 0.04 (±0.04) | 0.00 (±0.00) | 0.05 (±0.06) | 0.05 (±0.07) | 0.02 (±0.02) |
| PointReach-random | 44.30 (±0.24) | 30.80 (±1.74) | 7.67 (±1.97) | 1.59 (±0.52) | 1.37 (±0.09) | 1.78 (±0.14) | 1.52 (±0.26) | 33.90 (±3.28) | **45.17** (±0.13) | **45.40** (±0.16) |
| PointRooms-random | 35.52 (±0.80) | 24.10 (±0.81) | 4.67 (±0.80) | 1.44 (±0.77) | 1.43 (±0.18) | 1.61 (±0.17) | 1.29 (±0.37) | 29.74 (±3.37) | **36.16** (±1.16) | **36.23** (±0.79) |
| Reacher-random | **41.12** (±0.11) | 22.52 (±0.77) | 15.35 (±1.95) | 3.96 (±0.50) | 1.66 (±0.30) | 2.52 (±0.28) | 2.54 (±0.17) | 10.56 (±1.36) | 34.48 (±8.12) | 32.92 (±5.42) |
| SawyerReach-random | **41.05** (±0.19) | 14.86 (±3.27) | 11.30 (±2.12) | 2.99 (±1.32) | 0.58 (±0.21) | 1.36 (±0.14) | 1.18 (±0.29) | 29.09 (±4.23) | **39.27** (±2.16) | **41.86** (±0.55) |
| SawyerDoor-random | 36.82 (±3.20) | 25.86 (±1.12) | 25.33 (±1.46) | 3.59 (±0.64) | 3.73 (±0.83) | 9.82 (±1.08) | 4.36 (±0.86) | 29.31 (±4.10) | 28.85 (±1.99) | 28.66 (±3.53) |
| FetchReach-random | **46.50** (±0.09) | 38.26 (±0.24) | 30.86 (±8.49) | 2.31 (±1.60) | 0.84 (±0.31) | 0.19 (±0.04) | 0.97 (±0.23) | 40.86 (±2.65) | **47.01** (±0.07) | **47.12** (±0.04) |
| FetchPush-random | **11.48** (±1.03) | 5.01 (±0.64) | 3.01 (±0.71) | 3.85 (±0.17) | 3.14 (±0.25) | 3.60 (±0.42) | 3.67 (±0.65) | 5.14 (±1.55) | 5.65 (±0.64) | **5.79** (±1.16) |
| FetchSlide-random | **2.34** (±0.28) | 1.47 (±0.12) | 0.19 (±0.11) | 0.13 (±0.13) | 0.25 (±0.13) | 0.20 (±0.29) | 0.15 (±0.09) | 0.50 (±0.50) | 0.59 (±0.30) | 0.55 (±0.08) |
| FetchPick-random | **8.06** (±0.71) | 3.05 (±0.42) | 2.01 (±0.46) | 1.64 (±0.30) | 1.84 (±0.17) | 1.84 (±0.58) | 1.73 (±0.17) | 2.56 (±1.13) | 3.91 (±1.29) | 4.03 (±0.56) |
| HandReach-random | **5.23** (±0.55) | **3.96** (±0.81) | 0.00 (±0.00) | 0.00 (±0.00) | 0.00 (±0.00) | 0.00 (±0.00) | 0.00 (±0.00) | 0.01 (±0.02) | 0.00 (±0.01) | 0.00 (±0.00) |

