# OpenReview forum: "Rethinking Goal-Conditioned Supervised Learning and Its Connection to Offline RL"
_ICLR.cc/2022/Conference — ICLR 2022 Poster_

### Official Review · Reviewer_fo1Z · 2021-11-01

**Correctness:** 3
**Technical Novelty And Significance:** 2
**Empirical Novelty And Significance:** 3
**Recommendation:** 6
**Confidence:** 3

**Main Review:**

Strengths
- The paper proposes a weighted GCSL approach that appears to be motivated from optimizing a better lower bound (Theorem 1) and results in large empirical gains (although statistical uncertainty in results isn’t reported).
- The empirical experiments are thorough with ablations and show the benefit of the proposed WCGSL.
- The paper is generally positioned well wrt prior work but more discussion of some closely related work is needed for clarity.

Weaknesses:

- I found the paper to be somewhat unclear to read. Some examples:
  - The main contributions are presented very succinctly in Section 3.2. This makes the (1) goal-conditioned advantage re-weighting and (2) best-advantage weight hard to comprehend and understand the motivation behind such schemes.
  - Notations are overloaded heavily, for example, the advantage and value functions are used without any superscript for the policy which was confusing. For example, I didn’t understand what value function is being talked about in Proposition 2.
  - Results from Wang et. al (2018) and Haarnoja et al (2017) are directly used without clearly explaining them to the reader.
Multi-modality challenge is used in intro and section 3.2 without introducing what is the challenge (it only becomes clear in the last part of sec 3.2)
  - The motivating example was confusing – if we only have a sparse 0/1 reward function for goal reaching (as defined in the preliminaries), how are we claiming one trajectory to be better than another? Under this reward function, both the trajectories achieve the same return.

- Theoretical presentation of some results don’t add much value. Examples:
  - Theorem 2  trivially follows from Theorem 1 following a 2 line proof and calling it a theorem is unwarranted.
  - Proposition 2 is presented in the paper but empirical experiments simply avoid its relabeling strategy and implications. This made me wonder about the relevance of that proposition for this paper.


- The empirical reporting and rigor of the results should be improved.
   - No error bars are reported in Table 1 but as shown by Agarwal et al. (2021), it seems critical to report statistical uncertainty in deep RL results. Furthermore, the large number of entries in Table 1 makes it slightly overwhelming to read, maybe presenting it visually using performance profiles might be more informative.
  - The learning curves are only shown selectively for tasks where WGCSL outperforms other methods. Instead, curves showing aggregate scores across all tasks such as mean scores or interquartile mean (less prone to outliers)  for random and expert datasets with confidence intervals (CIs) might be a better way to summarize performance across tasks.
  - Adding average probability of improvement of WGCSL over other methods as done by Cong et al (2021) would further make the results more statistically reliable. The rliable library would be quite useful for reporting results with CIs and aggregate metrics: https://github.com/google-research/rliable
  - No std are shown for some of the methods in Figure 5.
  - Why are the default hyper parameters tuned for behavior regularization methods such as CQL and BCQ – it seems unlikely that the default values from D4RL/Mujoco tasks would work for the tasks presented here. Thus, are the comparisons fair?

Other minor comments:
 - Typo: GSCL → GCSL (under expert datasets in experimental section)


References:

[1] Haarnoja, T., Tang, H., Abbeel, P. and Levine, S., 2017, July. Reinforcement learning with deep energy-based policies. In International Conference on Machine Learning (pp. 1352-1361). PMLR.

[2] Wang, Q., Xiong, J., Han, L., Sun, P., Liu, H. and Zhang, T., 2018, December. Exponentially Weighted Imitation Learning for Batched Historical Data. In NeurIPS.

[3] Agarwal, R., Schwarzer, M., Castro, P.S., Courville, A. and Bellemare, M.G., 2021. Deep reinforcement learning at the edge of the statistical precipice. In NeurIPS.


**Summary Of The Paper:**

The paper proposes weighted goal-conditioned supervised learning (WGCSL) as a tighter lower bound to optimize than the typical GCSL objective. Then the paper combines a bunch of prior approaches / heuristics to set these weights and shows improvements on a new offline RL benchmark containing various simulated robotic manipulation tasks.

**Summary Of The Review:**

While the empirical results of the proposed approach looks promising, as pointed in the weaknesses, I have concerns about clarity of the paper, presentation of the theory and empirical rigor. If the authors can address these issues, I'd be willing to re-assess the paper.

--- After rebuttal --- Updated my score to 6.

---

> ### Author Response · Authors · 2021-11-17
> **Response to reviewer fo1Z (part 1/3)**
>
> Thanks for your thoughtful comments. We provide clarification to your questions as below. We appreciate it if you have any further questions or comments.
>
> **Q1**: The paper proposes a weighted GCSL approach that appears to be motivated from optimizing a better lower bound (Theorem 1) and results in large empirical gains (although statistical uncertainty in results isn’t reported).
>
> **A1**: Thanks for pointing this out. We have provided updated results with statistical uncertainty in Appendix E.1 (page 18) and Table
> 4 (page 29). The results show that WGCSL consistently outperforms other algorithms, and it is as stable as supervised methods such as GCSL.
>
> **Q2**: The main contributions are presented very succinctly in Section 3.2. This makes the (1) goal-conditioned advantage re-weighting and (2) best-advantage weight hard to comprehend and understand the motivation behind such schemes.
>
> **A2**: Thanks for the feedback. We have revised our paper and included additional discussions about the motivation behind those weights. Specifically, goal-conditioned advantage re-weighting (GEAW) is inspired by Corollary 1 that we can reweight samples using certain functions to improve the performance.
> Using a learned value function is a natural choice for evaluating the importance of samples, which was also adopted by prior works, e.g., AWR, AWAC, and MARWIL. The intuition behind GEAW is that we are assigning larger weights for samples with higher values/advantages so that the agent is inclined to learn more from "better" samples.
>
> The best-advantage weight (BAW) is introduced to tackle the multimodality problem, where, in general, there are many valid trajectories from state $s_t$ to the same goal $g$. This problem causes multiple counteracting action labels, which can impede learning. In order to handle this problem, we introduce the best advantage weight in our general weighting scheme to better fit the distribution with the highest return.
>
> **Q3**: Notations are overloaded heavily, for example, the advantage and value functions are used without any superscript for the policy which was confusing. For example, I didn’t understand what value function is being talked about in Proposition 2.
>
> **A3**: In Proposition 2, we discuss the relationship between the expected values of the induced policy from the relabeled dataset and from the original dataset. Under certain conditions, the expected value of the induced policy from the relabeled dataset is larger than that of the original dataset, $\forall s\_t, g, s\_i \in \mathcal{D}\_{\tau}, \mathbb{E}\_{\tau \sim \pi\_b, s\_t,s\_i \sim \tau}\[V^{\pi\_{relabel}}(s\_t,\phi(s\_i))\] \geq \mathbb{E}\_{\tau \sim \pi\_b, s\_t,g \sim \tau}\[V^{\pi\_b}(s\_t,g)\]$. Here the policy $\pi\_b$ is the behaviour policy generating the dataset in offline RL settings. We have added this clarification in the revision.
>
> **Q4**: Results from Wang et al. (2018) and Haarnoja et al. (2017) are directly used without clearly explaining them to the reader. Multi-modality challenge is used in intro and section 3.2 without introducing what is the challenge (it only becomes clear in the last part of sec 3.2)
>
> **A4**: Thanks for the feedback to improve the presentation. We have included additional discussions about the inspiration of Wang et al. (2018) and Haarnoja et al. (2017) in our revised version. The multimodality problem in goal-conditioned RL is that, in general, there are many valid trajectories from the same state to the same goal. The distribution over actions can be highly multimodal. This problem posts challenges of multiple counteracting action labels, which can impede learning [1]. We have refined the introduction in our revision.
>
> [1] Lynch, Corey, et al. "Learning latent plans from play." Conference on Robot Learning. PMLR, 2020.
>
> **Q5**: The motivating example was confusing – if we only have a sparse 0/1 reward function for goal reaching (as defined in the preliminaries), how are we claiming one trajectory to be better than another? Under this reward function, both the trajectories achieve the same return.
>
> **A5**: In the preliminaries, we have introduced the goal-conditioned reinforcement learning setting, in which the objective is to maximize the discounted cumulative return. In this setting, if we reach the goal faster and stay at the goal longer, we can obtain a higher return. In our motivating example, WGCSL selects the shortest route, while GCSL chooses a tortuous one. Therefore, the cumulative return of WGCSL is higher than GCSL.
>
> **Q6**: Theorem 2 trivially follows from Theorem 1 following a 2 line proof and calling it a theorem is unwarranted.
>
> **A6**: Thanks. We have updated it as a corollary.

---

> > ### Author Response · Authors · 2021-11-17
> > **Response to reviewer fo1Z (part 2/3)**
> >
> > **Q7**: Proposition 2 is presented in the paper but empirical experiments simply avoid its relabeling strategy and implications. This made me wonder about the relevance of that proposition for this paper.
> >
> > **A7**: Actually the relabeling strategy of Proposition 2 works equally well as our empirical strategy, except for a longer training time. We provide an additional comparison between the relabeling strategy of Proposition 2 (called WGCSL-slow-relabel) and the relabeling strategy that we used in the following table. The reason we use a slightly different relabeling strategy is in consideration of its simplicity and efficiency in implementation. The relabeling strategy of Proposition 2 requires a complete scan of all future states of the trajectory, which is costly. We find that relabeling with random future states has a similar performance empirically.
> >
> > |                   | WGCSL             | WGCSL-slow-relabel   |
> > |-------------------|-------------------|----------------------|
> > | PointRooms-expert | 36.15 ($\pm$0.85) | 37.08 ($\pm$0.56)    |
> > | FetchReach-expert | 46.33 ($\pm$0.04) | 46.15 ($\pm$0.10)    |
> > | FetchPush-expert  | 39.11 ($\pm$0.17) | 39.26 ($\pm$0.28)    |
> > | FetchSlide-expert | 10.73 ($\pm$1.09) | 9.34 ($\pm$1.71)     |
> > | FetchPick-expert  | 34.37 ($\pm$0.51) | 33.99 ($\pm$1.81)    |
> > | HandReach-expert  | 26.73 ($\pm$1.20) | 28.61 ($\pm$1.80)    |
> > | PointRooms-random | 35.52 ($\pm$0.80) | 36.56 ($\pm$1.28)    |
> > | FetchReach-random | 46.50 ($\pm$0.09) | 46.61 ($\pm$0.07)    |
> > | FetchPush-random  | 11.48 ($\pm$1.03) | 11.64 ($\pm$2.07)    |
> > | FetchSlide-random | 2.34 ($\pm$0.28)  | 1.81 ($\pm$0.41)     |
> > | FetchPick-random  | 8.06 ($\pm$0.71)  | 6.48 ($\pm$2.24)     |
> > | HandReach-random  | 5.23 ($\pm$0.55)  | 5.29 ($\pm$1.71)     |
> >
> > **Q8**: No error bars are reported in Table 1 but as shown by Agarwal et al. (2021), it seems critical to report statistical uncertainty in deep RL results. Furthermore, the large number of entries in Table 1 makes it slightly overwhelming to read, maybe presenting it visually using performance profiles might be more informative.
> >
> > **A8**: Thanks for the suggestions. We have provided updated results with statistical uncertainty in Appendix E.1 and Table 4 (page 29). Also, we presented Table 1 visually in Appendix E.1 and Figure 7. The results show that WGCSL consistently outperforms other algorithms, and it is as stable as supervised methods such as GCSL.
> >
> > **Q9**: The learning curves are only shown selectively for tasks where WGCSL outperforms other methods. Instead, curves showing aggregate scores across all tasks such as mean scores or interquartile mean (less prone to outliers) for random and expert datasets with confidence intervals (CIs) might be a better way to summarize performance across tasks.
> >
> > **A9**: We show a subset of all results in the main page because they are the hardest tasks in our experiments. In fact, we also provided the learning curves of all experiments in Appendix E.1 and Figure 7, which consistently show that our method WGCSL performs better or comparably with baselines. Following the reviewer's suggestion, we also provide an aggregate result across all environments in Appendix E.3 and Figure 9.
> >
> > **Q10**: Adding average probability of improvement of WGCSL over other methods as done by Cong et al (2021) would further make the results more statistically reliable. The rliable library would be quite useful for reporting results with CIs and aggregate metrics: https://github.com/google-research/rliable
> >
> > **A10**: Thanks for your suggestion. We provide the normalized score across 10 environments in Figure 9, Appendix E.3. We can observe the significant improvement of WGCSL over other baselines. We would like to try the package you mentioned as an additional evaluation method when time permits.
> >
> > **Q11**: No std are shown for some of the methods in Figure 5.
> >
> > **A11**: In fact, we do plot the std for all algorithms in Figure 5. However, the std of some algorithms are so small that we may not notice them at first glance, especially in PointReach and FetchReach environments. The reason is that those two environments are easier to train. Actually, if you zoom in, you can find there is still a small light std area at the start of the training curve in those environments. Moreover, we provide our source code with the offline datasets, and you can check the small standard deviation in these tasks.

---

> > > ### Author Response · Authors · 2021-11-17
> > > **Response to reviewer fo1Z (part 3/3)**
> > >
> > > **Q12**: Why are the default hyper parameters tuned for behavior regularization methods such as CQL and BCQ – it seems unlikely that the default values from D4RL/Mujoco tasks would work for the tasks presented here. Thus, are the comparisons fair?
> > >
> > > **A12**: We conduct a grid search across the four hardest tasks on critical hyperparameters of CQL and BCQ, and report the results in the following table. For CQL, we vary the Lagrange threshold $\tau = \[2.0,5.0,10.0\]$, as suggested in tuning guidelines of the original paper [2]. For BCQ, we vary the permutation parameter $\Phi = \[0.1,0.05,0.01\]$ and batch size $bs = \[512,256,128\]$. We observe that in CQL the default hyperparameter performs best in almost all tasks, while in BCQ different hyperparameters achieve similar results. Given that WGCSL significantly outperforms CQL and BCQ under all hyperparameters configurations, it is safe to claim that we conduct fair comparisons. In fact, sparse reward tasks are generally challenging for model-free offline algorithms, as reported in prior works [3]. Generally, algorithms that are designed for offline goal-conditioned RL, such as WGCSL, HER, and AM, perform better than CQL and BCQ, as shown in Table 4 (Page 29).
> > >
> > > | CQL               | $\tau$ 2.0       | $\tau$ 5.0\*       | $\tau$ 10.0      |
> > > | :---------------- | :--------------- | :--------------- | :--------------- |
> > > | FetchPick-expert  | 2.95 ($\pm$0.59) | 3.28 ($\pm$0.56) | 1.68 ($\pm$0.18) |
> > > | FetchPush-expert  | 2.83 ($\pm$0.58) | 4.83 ($\pm$0.50) | 3.17 ($\pm$0.48) |
> > > | FetchSlide-expert | 0.29 ($\pm$0.06) | 0.72 ($\pm$0.09) | 0.10 ($\pm$0.08) |
> > > | HandReach-expert  | 0.00 ($\pm$0.00) | 0.00 ($\pm$0.00) | 0.00 ($\pm$0.00) |
> > > | FetchPick-random  | 1.88 ($\pm$0.27) | 1.88 ($\pm$0.37) | 1.54 ($\pm$0.16) |
> > > | FetchPush-random  | 3.33 ($\pm$0.82) | 3.75 ($\pm$0.27) | 3.12 ($\pm$0.27) |
> > > | FetchSlide-random | 0.04 ($\pm$0.06) | 0.12 ($\pm$0.00) | 0.33 ($\pm$0.22) |
> > > | HandReach-random  | 0.00 ($\pm$0.00) | 0.00 ($\pm$0.00) | 0.00 ($\pm$0.00) |
> > >
> > > | BCQ               | $\Phi$0.1 bs512  | $\Phi$0.1 bs256  | $\Phi$0.1 bs128  | $\Phi$0.05 bs512 | $\Phi$0.05 bs256 | $\Phi$0.05 bs128\* | $\Phi$0.01 bs512 | $\Phi$0.01 bs256 | $\Phi$0.01 bs128 |
> > > | :---------------- | :--------------- | :--------------- | :--------------- | :--------------- | :--------------- | :--------------- | :--------------- | :--------------- | :--------------- |
> > > | FetchPick-expert  | 2.01 ($\pm$0.73) | 2.26 ($\pm$1.02) | 2.25 ($\pm$0.61) | 2.09 ($\pm$0.57) | 1.88 ($\pm$0.12) | 2.00 ($\pm$0.54) | 2.15 ($\pm$0.09) | 1.77 ($\pm$0.38) | 1.27 ($\pm$0.56) |
> > > | FetchPush-expert  | 3.08 ($\pm$0.52) | 3.42 ($\pm$0.72) | 3.36 ($\pm$0.74) | 4.33 ($\pm$0.77) | 4.83 ($\pm$1.50) | 2.88 ($\pm$0.47) | 3.63 ($\pm$1.04) | 4.33 ($\pm$0.31) | 4.17 ($\pm$0.60) |
> > > | FetchSlide-expert | 0.29 ($\pm$0.21) | 0.23 ($\pm$0.11) | 0.37 ($\pm$0.18) | 0.18 ($\pm$0.12) | 0.21 ($\pm$0.04) | 0.21 ($\pm$0.16) | 0.27 ($\pm$0.09) | 0.25 ($\pm$0.15) | 0.25 ($\pm$0.03) |
> > > | HandReach-expert  | 0.12 ($\pm$0.04) | 0.11 ($\pm$0.05) | 0.15 ($\pm$0.08) | 0.11 ($\pm$0.06) | 0.04 ($\pm$0.01) | 0.04 ($\pm$0.03) | 0.00 ($\pm$0.00) | 0.02 ($\pm$0.02) | 0.00 ($\pm$0.00) |
> > > | FetchPick-random  | 1.81 ($\pm$0.34) | 1.98 ($\pm$0.57) | 2.40 ($\pm$0.43) | 1.61 ($\pm$0.11) | 1.98 ($\pm$0.86) | 2.54 ($\pm$0.88) | 1.97 ($\pm$0.17) | 1.99 ($\pm$0.71) | 2.17 ($\pm$0.54) |
> > > | FetchPush-random  | 3.15 ($\pm$0.65) | 2.26 ($\pm$0.59) | 2.78 ($\pm$1.14) | 3.87 ($\pm$0.44) | 2.34 ($\pm$0.68) | 3.49 ($\pm$1.19) | 1.87 ($\pm$0.27) | 2.02 ($\pm$0.51) | 3.37 ($\pm$0.77) |
> > > | FetchSlide-random | 0.08 ($\pm$0.12) | 0.01 ($\pm$0.02) | 0.19 ($\pm$0.27) | 0.20 ($\pm$0.22) | 0.17 ($\pm$0.12) | 0.25 ($\pm$0.20) | 0.08 ($\pm$0.12) | 0.08 ($\pm$0.12) | 0.08 ($\pm$0.12) |
> > > | HandReach-random  | 0.00 ($\pm$0.00) | 0.00 ($\pm$0.00) | 0.00 ($\pm$0.00) | 0.00 ($\pm$0.00) | 0.00 ($\pm$0.00) | 0.00 ($\pm$0.00) | 0.00 ($\pm$0.00) | 0.00 ($\pm$0.00) | 0.00 ($\pm$0.00) |
> > >
> > > \*: the default hyperparameter.
> > >
> > > [2] Kumar, Aviral, et al. "Conservative q-learning for offline reinforcement learning." arXiv preprint arXiv:2006.04779 (2020).
> > >
> > > [3] Fu, Justin, et al. "D4rl: Datasets for deep data-driven reinforcement learning." arXiv preprint arXiv:2004.07219 (2020).
> > >
> > > **Q13**: Typo: GSCL → GCSL (under expert datasets in experimental section)
> > >
> > > **A13**: Thanks for pointing out the typos. We have fixed them in our revised manuscript.

---

> > > > ### Comment · Reviewer_fo1Z · 2021-11-17
> > > > **Thanks for the response!**
> > > >
> > > > Dear authors,
> > > >
> > > > Thanks for the updates for addressing my concerns and making the paper clearer. Assuming that you would report the statistical uncertainty in results in the main paper (and not just the appendix) and substantiate your claims of significant improvements with either statistical significance tests or metrics like [average probability of improvement](https://github.com/google-research/rliable#probability-of-improvement), I have updated my score from 5 to 6.

---

### Official Review · Reviewer_FPSj · 2021-11-02

**Correctness:** 4
**Technical Novelty And Significance:** 3
**Empirical Novelty And Significance:** 3
**Recommendation:** 8
**Confidence:** 3

**Main Review:**

Pros:
The paper is well written and the notation very easy to follow.  There are a couple of typos/weird phrase constructions in the paper that could warrant a bit of proofreading.  I put some examples at the end of this section.
The idea, although somewhat an obvious followup to GCSL, is well explained and analyzed, and experimental results are convincing.

Cons:
One of the advantages of GCSL is that it is a purely supervised method, which greatly enhances the simplicity and stability.  The proposed approach uses a Q-function to derive the advantage function A, which requires a bootstrap-based method to be applied to an offline dataset.  I would greatly appreciate a bit of commenting on this in the paper, as this is well-known to not be trivial to get working.  Is the method relying on the goal-conditioned aspect to compensate for over-estimation issues?

Another topic I would have like to see more discussion on is the difference between HER and WGCSL.  If I understand correctly, HER is able to train in the offline setting due to goal-conditioned aspects of the problem, but why would WGCSL fundamentally work better than a purely value-based method?

Overall I vote to accept this paper, but would appreciate responses from the authors on my questions.

Section: 'Expert Datasets': 'Tabel 1' -> 'Table 1'.  'in offline' -> 'in the offline'
'Random Datasets' section: 'relatively well policy'  -> 'relatively good policy'.  'results of the random' -> 'results on the random'.

**Summary Of The Paper:**

This paper proposes an extension of GCSL where the goal-conditioned BC loss is weighted by a variable that correlates with the number of steps necessary to achieve the desired goal.  The advantage of this approach is that sub-optimal trajectories to a particular goal will get downweighted to the benefit of more direct trajectories.  This effectively adds a policy improvement step to GCSL with the underlying objective function being that of arriving at goal states as fast as possible.

Results on goal-conditioned tasks show that the proposed approach (WGCSL) performs consistently better than GCSL and in some cases significantly better (HandReach-expert).

**Summary Of The Review:**

A somewhat obvious extension to GCSL (online goal-conditioned BC), but that is well-explained and shows convincing experimental results.  I vote for acceptance, and believe this provides a potentially new baseline for offline goal-based methods.

---

> ### Author Response · Authors · 2021-11-17
> **Response to reviewer FPSj**
>
> Thanks for your inspiring comments. We provide clarification to your questions as below. We appreciate it if you have any further questions or comments.
>
> **Q1**: One of the advantages of GCSL is that it is a purely supervised method, which greatly enhances the simplicity and stability. The proposed approach uses a Q-function to derive the advantage function A, which requires a bootstrap-based method to be applied to an offline dataset. I would greatly appreciate a bit of commenting on this in the paper, as this is well-known to not be trivial to get working. Is the method relying on the goal-conditioned aspect to compensate for over-estimation issues?
>
> **A1**: Some offline RL algorithms that use a value/Q-function usually involve a maximization operator over the value/Q-function, which causes over-estimation issues. However, in our algorithm, the value function is used as a weight to evaluate the importance of different samples in the dataset. Essentially it reweights the distribution of the dataset. The trained policy does not try to maximize the Q-value, but to mimic the resampled experiences of the dataset. Theoretically, WGCSL is similar to AWAC [1] and MARWIL, which introduce implicit policy constraints to alleviate the out-of-distribution problem. Therefore, our algorithm naturally alleviates over-estimation issues. As shown in our Figure 6, the estimated value of WGCSL is rather small compared to HER and DDPG algorithms in the HandReach task.
>
> [1] Nair A, Dalal M, Gupta A, et al. AWAC: Accelerating Online Reinforcement Learning with Offline Datasets.
>
> **Q2**: Another topic I would have like to see more discussion on is the difference between HER and WGCSL. If I understand correctly, HER is able to train in the offline setting due to goal-conditioned aspects of the problem, but why would WGCSL fundamentally work better than a purely value-based method?
>
> **A2**: As the reviewer expected, the multi-goal part, e.g., goal relabeling can slightly alleviate the offline problem via data augmentation as shown in Appendix E.2 and Figure 8. However, HER learns a policy to maximize the value function, which can deviate far from the data distribution and has severe over-estimation issues in offline settings. Figure 6 shows the overestimation problem of HER, i.e., the estimated value of HER is very large, which results in poor performance. In contrast, WGCSL learns the value function to reweight the samples in the dataset. Therefore our algorithm fundamentally works better than a purely value-based method.
>
> **Q3**: Section: 'Expert Datasets': 'Tabel 1' -> 'Table 1'. 'in offline' -> 'in the offline' 'Random Datasets' section: 'relatively well policy' -> 'relatively good policy'. 'results of the random' -> 'results on the random'.
>
> **A3**: Thanks for spotting the typos. We have corrected them in the revision.

---

### Official Review · Reviewer_i8oK · 2021-11-02

**Correctness:** 2
**Technical Novelty And Significance:** 3
**Empirical Novelty And Significance:** 3
**Recommendation:** 5
**Confidence:** 3

**Main Review:**

**Significance**: On the one hand, the empirical results of the paper are quite strong. On the other hand, by introducing a value function, the paper significantly increases the complexity of the method, and simplicity is arguably the most important feature of goal-conditioned behavior cloning methods.

**Originality**: The proposed method is novel, to the best of my knowledge. While both $\gamma$-discounting and advantage weighting have been studied in a number of prior works (as noted in this paper), I am unaware of prior work that applies them to goal-conditioned behavior cloning methods.

**Clarity**: The paper is clear and readable.
* I was confused about the difference between GoalBC, Goal BCQ, and Goal CQL. For example, is Goal-CQL designed so that Q function reflects the *marginal* data collection policy or the *goal-conditioned* data collection policy? (i.e., how is hindsight relabeling combined with CQL)?

**Correctness**: I have a few concerns about the correctness of the paper.
* In the second to last line of the proof, if we replaced the weights $\gamma^{i-t}$ with 0, we'd get a useless objective, but would the bound be made *even* tighter?
* I'm unsure if the main result is actually useful. In particular, Theorem 1 shows that the proposed method optimizes a tighter bound on $J_{surr}$, but we actually want a tighter bound on $J$ (defined in Sec. 2).
* Proposition 1, "good as or better than" -- According to what criterion?
* Proposition 2: I think this result is a bit misleading, because it is really a result about the dataset, not the policy; it is not a "policy improvement" result, as indicated by the subsection header.

**Minor comments**

* Abstract: It's not clear what problem is being solved. I'd recommend making this more clear in the first 1-3 sentences.
* "can we learn goal-conditioned policies from offline data?" -- [1] already does this.
* "hindsight relabeling would generate non-optimal experiences" -- I don't this this is correct: [2] prove that hindsight relabeling is equivalent to inferring a task assuming that the experience were optimal.
* "extremely sparse" -> "sparse"
* "last step reward" -- Clarify what this means.
* Eq. 2, $w_{i,t}$: This should be defined before the equation.
* I'd recommend including advantage weighting methods (e.g., [3]) as an additional baseline. This should be trivial to implement because the proposed method already includes a form of advantage weighting.
* Ethics Statement, "there are concerns that AGI might go out of human control in the future" -- The ethics statement lacks nuance. I'd recommend that it be revised to more carefully address the specific ethical implications of this paper.


[1] Lynch, Corey, et al. "Learning latent plans from play." Conference on Robot Learning. PMLR, 2020.

[2] Eysenbach, Benjamin, et al. "Rewriting history with inverse rl: Hindsight inference for policy improvement." arXiv preprint arXiv:2002.11089 (2020).

[3] Peng, Xue Bin, et al. "Advantage-weighted regression: Simple and scalable off-policy reinforcement learning." arXiv preprint arXiv:1910.00177 (2019).

**Summary Of The Paper:**

The proposed method proposes a method for goal-conditioned RL that can be interpreted as a weighted version of prior work.These weights resemble a combination of discounting and advantage weighting. The paper provides theory arguing that these weights cause the proposed method to optimize a tighter lower bound than prior work. Experiments show that it outperforms the prior work, which does not include weights on each training example.

**Summary Of The Review:**

My main concerns are about the clarity and correctness of the paper (the empirical results are already great!). If the paper were revised to address these concerns, I'd be inclined to change my vote.

---

> ### Author Response · Authors · 2021-11-17
> **Response to reviewer i8oK (part 1/3)**
>
> Thanks for your constructive comments. We provide clarification to your questions and concerns as below. We appreciate it if you have any further questions or comments.
>
> **Q1**: On the other hand, by introducing a value function, the paper significantly increases the complexity of the method, and simplicity is arguably the most important feature of goal-conditioned behavior cloning methods.
>
> **A1**: Due to the use of the value function, WGCSL is slightly more complex than GCSL. However, we theoretically and empirically show that this additional complexity is worthy of achieving significant performance improvement. Theoretically, the introduced value function enables WGCSL to learn a uniformly better policy than GCSL. Empirically, in the following table and our Appendix E.4, we perform an additional comparison of the training time between those algorithms, showing that WGCSL introduces very little extra training time on top of GCSL. Considering the large improvement brought by WGCSL in offline goal-conditioned RL, we believe that the cost of learning the value function is worthwhile.
>
> | Training Time (min)     | GCSL   |  WGCSL  | HER  | AM   |
> | ----------- | ----------- | ----------   |  -------    |  --------    |
> | FetchPush    |   53.82 $\pm$ 0.39    | 58.183 $\pm$ 0.95 | 59.86 $\pm$ 0.37  |  85.56 $\pm$ 0.94 |
> | FetchSlide   |    57.19 $\pm$ 0.24    |  58.47 $\pm$ 0.56   | 61.75 $\pm$ 0.98  | 85.97 $\pm$ 1.09  |
> |  FetchPick   |  54.26 $\pm$ 0.41  | 57.78 $\pm$ 0.44 | 59.91 $\pm$ 0.54 | 102.53 $\pm$ 1.28 |
> | HandReach   | 51.20 $\pm$ 0.38  | 55.98 $\pm$ 0.59 | 56.47 $\pm$ 0.75 | 109.62 $\pm$ 0.96 |
>
> **Q2**: I was confused about the difference between Goal BC, Goal BCQ, and Goal CQL. For example, is Goal CQL designed so that Q function reflects the marginal data collection policy or the goal-conditioned data collection policy? (i.e., how is hindsight relabeling combined with CQL)?
>
> **A2**: We can extend BC, BCQ, and CQL to goal-conditioned settings by concatenating observations, desired goals, and achieved goals as states such that the Q function is goal-dependent. We design the Q function to reflect the goal-conditioned data collection policy. We refine the implementation details of those algorithms in appendix D.
>
> We did not combine hindsight relabeling with CQL in our experiments. However, to combine CQL with HER, we can follow a recent work, Actionable Models (AM) [1]. The combination can be considered as two stages: 1) hindsight relabeling to augment the training data, 2) training goal-conditioned CQL on the augmented data. We have included the results of AM in Appendix E.2 (Figure 8) in our revised manuscript, from which we can observe the effectiveness of AM. In addition, WGCSL is more efficient than AM and requires less training time.
>
> [1] Chebotar Y, Hausman K, Lu Y, et al. Actionable Models: Unsupervised Offline Reinforcement Learning of Robotic Skills. ICML 2021.
>
> **Q3**: In the second to last line of the proof, if we replaced the weights $\gamma^{i-t}$ with $0$, we'd get a useless objective, but would the bound be made \emph{even} tighter?
>
> **A3**: No, it wouldn't. Theorem 1 assumes $\pi$ to be a random discrete policy, the surrogate function $J\_{surr}(\pi) \leq 0$. Therefore replacing $\gamma^{i-t}$ with $0$ results in an upper bound rather than a lower bound of $J\_{surr}(\pi)$.
>
> **Q4**: I'm unsure if the main result is actually useful. In particular, Theorem 1 shows that the proposed method optimizes a tighter bound on $J\_{surr}$, but we actually want a tighter bound on $J$ (defined in Sec. 2).
>
> **A4**:
> Theorem 1 is useful. We prove that the surrogate function $J\_{surr}$ has the same gradient on $\pi\_b$ with the actual objective $J$. The proof is included in Appendix B.1 Remark. Therefore, taking a gradient update with a proper step size on $J\_{surr}$ can also improve the policy on the objective $J$. Practically, we implicitly constraint the KL divergence between $\pi$ and $\pi\_b$ with the goal-conditioned exponential advantage weight (GEAW) for policy improvement, i.e., optimizing $\log\pi(a|s,g) exp(A(s,g,a))$ is equivalent to solving this problem:
>
> $
> \pi = \arg\max\_{\pi}E\_{a\sim\pi}\[A(s,g,a)\],\quad D\_{KL}(\pi(\cdot|s,g)\|\pi\_{b}(\cdot|s,g))\leq \epsilon
> $,
>
> which is similarly introduced in AWAC [2].
>
> [2] Nair A, Dalal M, Gupta A, et al. AWAC: Accelerating Online Reinforcement Learning with Offline Datasets.

---

> > ### Author Response · Authors · 2021-11-17
> > **Response to reviewer i8oK (part 2/3)**
> >
> > **Q5**: Proposition 1, "good as or better than" -- According to what criterion?
> >
> > **A5**: The criterion is in terms of the value function $V^{\pi}(s\_t, \phi(s\_i))$. That is, the policy $\tilde{\pi}$ always yields larger value on any state-goal pair $(s\_t,\phi(s\_i)) \in D\_\tau, i \geq t$, than the relabeling policy $\pi\_{relabel}$, where $D\_\tau$ contains the experiences after goal relabeling.
> >
> > **Q6**: Proposition 2: I think this result is a bit misleading, because it is really a result about the dataset, not the policy; it is not a "policy improvement" result, as indicated by the subsection header.
> >
> > **A6**: Combining Proposition 1 and 2 shows the policy improvement result (as discussed in the last paragraph in Section 3.3). In Proposition 1, we have proved that our algorithm generates a uniformly better policy than the goal relabeling policy $\pi\_{relabel}$. Proposition 2 further illustrates that under certain conditions the relabeling policy has a larger expected return than the return of the behavior policy induced by the dataset. In offline settings, the dataset is assumed to be generated by some behavior policy $\pi\_b$. Combining Proposition 1 and 2, we can prove that our algorithm could generate a better policy than the behavior policy $\pi\_b$. As the goal relabeling process is part of our algorithm, we need this proposition to guarantee the learned policy can be better than the original offline dataset.
> >
> > On the other hand, our algorithm can also be used in online settings, in which case the policy that generates the dataset is the old policy $\tilde{\pi}\_{old}$. Combining Proposition 1 and 2 we have a "policy improvement" result, that is, $\mathbb{E}\[V^{\tilde{\pi}}(s\_t, \phi(s\_i))\] \geq \mathbb{E}\[V^{\tilde\pi\_{old}}(s\_t, \phi(s\_i))\]$.

---

> > > ### Author Response · Authors · 2021-11-17
> > > **Response to reviewer i8oK (part 3/3)**
> > >
> > > *For Minor comments*
> > >
> > > **Q7**: Abstract: It's not clear what problem is being solved. I'd recommend making this more clear in the first 1-3 sentences.
> > >
> > > **A7**: This paper focuses on the offline goal-conditioned RL problem, which requires learning to reach goals in the goal space utilizing previously collected data. We have refined the abstract in our revised version.
> > >
> > > **Q8**: "can we learn goal-conditioned policies from offline data?" -- [3] already does this.
> > >
> > > **A8**: Thanks for pointing out this relevant. We have refined this sentence and referred to [3] in the revision. [3] employs VAE to jointly learn a latent plan representation and goal/plan-conditioned policy, which addresses the multi-modality problem in offline settings. However, their method has the same problem as GCSL, as they assign uniform weights for all the experiences and may result in suboptimal policies.
> > >
> > > [3] Lynch, Corey, et al. "Learning latent plans from play." Conference on Robot Learning. PMLR, 2020.
> > >
> > > **Q9**: "hindsight relabeling would generate non-optimal experiences" -- I don't think this is correct: [4] prove that hindsight relabeling is equivalent to inferring a task assuming that the experience were optimal.
> > >
> > > **A9**: [4] proves that the final state of a trajectory of length $T$ is the 'optimal' relabeling goal when the reward function satisfies:
> > >
> > > $$r(s_t, a_t)=\begin{cases}-\infty,&\text{if }t=T\text{ and }s_t\neq g\cr 0,&\text{otherwise}\end{cases}$$
> > >
> > > which is not common in practice. Besides, it doesn't consider relabeling with all future states rather than only the last state.
> > >
> > > [4] Eysenbach B, Geng X, Levine S, et al. Rewriting history with inverse RL: Hindsight inference for policy improvement. NIPS 2020.
> > >
> > >
> > > **Q10**: "extremely sparse" -> "sparse"; Eq. 2, $w\_{i,t}$: This should be defined before the equation; "last step reward" -- Clarify what this means.
> > >
> > > **A10**: Thanks for your suggestions on the writing. We have made them more clear in the revision. The last step reward is $\mathbb{E}\_{g \sim p(g), \tau \sim \pi}\[r(s\_T, a\_T, g)\]$, while the cumulative reward is $\mathbb{E}\_{g \sim p(g), \tau \sim \pi} \[\sum\_{t=0}^T \gamma^t r(s\_t, a\_t, g)\]$.
> > >
> > > **Q11**: I'd recommend including advantage weighting methods (e.g., AWR) as an additional baseline. This should be trivial to implement because the proposed method already includes a form of advantage weighting.
> > >
> > > **A11**: We have included AWR as an additional baseline and added the results of AWR in Appendix E.1. The results show that WGCSL exceeds AWR in both final performance and learning efficiency.
> > >
> > > **Q12**: Ethics Statement, "there are concerns that AGI might go out of human control in the future" -- The ethics statement lacks nuance. I'd recommend that it be revised to more carefully address the specific ethical implications of this paper.
> > >
> > > **A12**: Thank you for the suggestion! We have improved our statements to be more specific in the revision.

---

> > > > ### Comment · Reviewer_i8oK · 2021-11-17
> > > > **Discussion**
> > > >
> > > > Thanks to the authors for addressing all these questions. Unless noted below, the answers and/or revisions have addressed my concerns! I have a few remaining questions:
> > > >
> > > > > We refine the implementation details of those algorithms in appendix D.
> > > >
> > > > These revisions make it much easier to understand the baselines; thanks! Could the authors also add the objective functions for GoalBCQ and GoalCQL? (I want to make sure I understand where the goals are being sampled from.)
> > > >
> > > > > replacing $\gamma^{i-t}$ with 0 results in an upper bound rather than a lower bound.
> > > >
> > > > I didn't understand this explanation (e.g., what does the policy being random have to do with this? where does the theorem assume that $J_\text{surr} \le 0$?)
> > > >
> > > > > offline setting
> > > >
> > > > Thanks for this revision. In the Abstract, I find the sentence "generalize it to the offline goal-conditioned RL problem" a bit disingenuous as many prior works (e.g., Lynch '20, Ding '19) have looked at this setting. I would recommend revising this sentence to clarify what the "theoretical foundation" is and how it's different from prior work.
> > > >
> > > > > hindsight relabeling would generate non-optimal experiences
> > > >
> > > > By definition, performing hindsight relabeling using inverse RL always generates optimal experience (see first section of Section 4 in [4], before Section 4.1). This result holds for any reward function. For most goal-conditioned reward functions, it *does* correspond to relabeling with all future states (as well as other states). In summary, Section 4 of [4] proves exactly the *opposite* of what the current paper claims it shows.
> > > >
> > > > Nit: The relevant sentence "However, hindsight ... discounted return" is a run-on sentence.

---

> > > > > ### Author Response · Authors · 2021-11-19
> > > > > **Further clarification on your questions (part 2/2)**
> > > > >
> > > > > **Q3**: In the Abstract, I find the sentence "generalize it to the offline goal-conditioned RL problem" a bit disingenuous as many prior works (e.g., Lynch '20, Ding '19) have looked at this setting. I would recommend revising this sentence to clarify what the "theoretical foundation" is and how it's different from prior work.
> > > > >
> > > > > **A3**: We would like to clear up a confusion. In the abstract, "Goal-conditioned Supervised Learning" specifically refers to the method in [3] rather than other prior works. As is introduced in the second paragraph of the Introduction, GCSL theoretically guarantees to optimize over a lower bound of the objective for goal reaching problem. Our main contribution is to generalize [3] to offline RL settings. We show that our method provides a tighter lower bound than GCSL, and guarantees monotonic policy improvement on offline datasets. We have refined the abstract to clarify this confusion.
> > > > >
> > > > > Thank the reviewer for pointing out additional references. Prior works [Lynch '20 and Ding '19] focus on imitation learning methods, which requires expert (play) demonstrations, while we are studying offline RL problem where the dataset is not required to be generated by an expert policy. In fact, our proposed WGCSL can even learn from randomly generated datasets. To be specific, Lynch '20 proposes to employ VAE to jointly learn a latent plan representation and goal/plan-conditioned policy. However, their method has the same problem as GCSL, as they assign uniform weights for all the experiences and may result in suboptimal policies. Ding '19 proposes to extend GAIL to goal-conditioned settings while combining hindsight relabeling to speed up training. Their method needs online interaction with the environment to train the discriminator, which is not possible in offline settings.
> > > > >
> > > > > **Q4**: By definition, performing hindsight relabeling using inverse RL always generates optimal experience (see the first section of Section 4 in [4], before Section 4.1). This result holds for any reward function. For most goal-conditioned reward functions, it does correspond to relabeling with all future states (as well as other states). In summary, Section 4 of [4] proves exactly the opposite of what the current paper claims it shows.
> > > > >
> > > > > Nit: The relevant sentence "However, hindsight ... discounted return" is a run-on sentence.
> > > > >
> > > > > **A4**: We would like to clarify that "hindsight relabeling" in our paper particularly refers to the strategy of $1(\psi=s\_T)$ introduced in Section 4.1 of HER [5], which is commonly used in goal-conditioned methods such as GCSL [6] and [7]. We have added this clarification in our claim (by mentioning "relabeling with actually reached states"). As [4] pointed out, $1(\psi=s\_T)$ is optimal when reward takes the form
> > > > > $$
> > > > > r(s\_t, a\_t)=\begin{cases}-\infty, &
> > > > > \text{if}\~t=T\~\text{and}\~s\_t\neq \psi
> > > > > \cr 0, &\text{otherwise}\end{cases}.
> > > > > $$
> > > > > In fact, if we replace $r$ with goal-conditioned RL reward
> > > > > $$
> > > > > r(s\_t, a\_t)=\begin{cases}1, &
> > > > > \Vert\phi(s\_t)-\psi\Vert\_2^2<\text{some threshold}
> > > > > \cr 0, &\text{otherwise}\end{cases}
> > > > > $$
> > > > > in Eq. 7 in [4]
> > > > > $$
> > > > > q(\psi|\tau)\propto p(\psi)\exp{\sum\_tr\_\psi(s\_t,a\_t)-\log Z(\psi)},
> > > > > $$
> > > > > we will see $q(\psi|\tau)\neq 1(\psi=s\_T)$. This is why we claim that 'hindsight relabeling' is generally non-optimal.
> > > > >
> > > > > [4] Eysenbach B, Geng X, Levine S, et al. Rewriting history with inverse RL: Hindsight inference for policy improvement. NIPS, 2020.
> > > > >
> > > > > [5] Andrychowicz M, Wolski F, Ray A, et al. Hindsight experience replay. NIPS 2017.
> > > > >
> > > > > [6] Ghosh D, Gupta A, Reddy A, et al. Learning to reach goals via iterated supervised learning. ICLR 2021.
> > > > >
> > > > > [7] Ding Y, Florensa C, Phielipp M, et al. Goal-conditioned imitation learning. NIPS, 2019.

---

> > > > > > ### Comment · Reviewer_i8oK · 2021-11-20
> > > > > > **Discussion**
> > > > > >
> > > > > > Q3: In the Abstract, I find the sentence "generalize it to the offline goal-conditioned RL problem" a bit disingenuous...
> > > > > >
> > > > > > My concern here isn't about whether the proposed method is different from these prior works (I agree with the authors about this). Rather, my concern is that the abstract seems to overclaim. From reaching the abstract, a reader would presume that this paper presents a radically new "framework" and that no prior paper has studied "offline goal-conditioned RL."
> > > > > >
> > > > > > Q4: By definition, performing hindsight relabeling using inverse RL always generates optimal experience
> > > > > >
> > > > > > My concern here is that the citation proves the *opposite* of what the current paper claims it does. This is misleading.

---

> > > > > > > ### Author Response · Authors · 2021-11-21
> > > > > > > **Further clarification on your questions**
> > > > > > >
> > > > > > > **A3:**
> > > > > > > Thanks for your further clarification. The sentence "generalize it to the offline goal-conditioned RL problem" in this context means that GCSL is an online algorithm maximizing last-step reward and we generalize GCSL to the offline goal-conditioned RL setting. We do not tend to claim that WGCSL is the first work for offline goal-conditioned RL problem, instead it provides a novel solution to this setting. We have made it more clear in the revision. We hope our revision can address your concern.
> > > > > > >
> > > > > > > **A4:**
> > > > > > > Thanks for your further clarification. We have revised our sentence in the Introduction to avoid this misleading. Please let us know if we have addressed your concern.
> > > > > > >
> > > > > > > We agree with the reviewer that "performing hindsight relabeling using inverse RL always generates optimal experience". However, this inverse-RL optimal relabeling strategy is computationally intractable and needs approximation. In our algorithm we stick with the widely used hindsight relabeling strategy as in GCSL[1] and HER[2], i.e., relabeling using future achieved states. As shown in our previous discussion, such relabeling strategy is simple and efficient, but in general is not optimal. Efficiently incorporating inverse RL relabeling strategy with WGCSL is indeed an interesting future direction.
> > > > > > >
> > > > > > > [1] Ghosh D, Gupta A, Reddy A, et al. Learning to reach goals via iterated supervised learning. ICLR 2021.
> > > > > > >
> > > > > > > [2] Andrychowicz M, Wolski F, Ray A, et al. Hindsight experience replay. NIPS 2017.

---

> > > > > > > > ### Comment · Reviewer_i8oK · 2021-11-22
> > > > > > > > **Discussion**
> > > > > > > >
> > > > > > > > > Abstract
> > > > > > > >
> > > > > > > > Thanks for the revision. I now think this part of the abstract is correct. (It is still pretty hard to read; shorter sentences may help).
> > > > > > > >
> > > > > > > > > Optimality of hindsight relabeling
> > > > > > > >
> > > > > > > > The first half of that sentence is now correct. Where does (Chebotar '2021) prove that relabeling with future states "would generate non-optimal experiences in more general offline goal-conditioned RL setting with cumulative discounted return"?

---

> > > > > > > > > ### Author Response · Authors · 2021-11-23
> > > > > > > > > **Response to the Discussion**
> > > > > > > > >
> > > > > > > > > Thanks for your feedback. We would like to provide explanation to your question.
> > > > > > > > >
> > > > > > > > > **Q1**: Thanks for the revision. I now think this part of the abstract is correct. (It is still pretty hard to read; shorter sentences may help).
> > > > > > > > >
> > > > > > > > > **A1**: Thanks for the advice. We have refined the relevant part with shorter sentences in the revision.
> > > > > > > > >
> > > > > > > > > **Q2**: The first half of that sentence is now correct. Where does (Chebotar '2021) prove that relabeling with future states "would generate non-optimal experiences in more general offline goal-conditioned RL setting with cumulative discounted return"?
> > > > > > > > >
> > > > > > > > > **A2**: We would like to clarify a potential confusion here. We refer [1] for the definition of cumulative discounted return $r(s\_t,a\_t)=\gamma^t\cdot 1[\phi(s_t)=g]$. The non-optimality of relabeling with future states is actually derived from [2], which has proven that the optimal relabelling strategy is
> > > > > > > > > $$
> > > > > > > > > q(g|\tau)\propto p(g)\exp{\sum\_tr(s\_t,a\_t)-\log Z(g)}=p(g)\exp{\sum\_t\gamma^t\cdot 1[\phi(s_t)=g]-\log Z(g)}.
> > > > > > > > > $$
> > > > > > > > > This is different from simply relabeling with future states as $q(g|\tau)=1[g=s_T]$ in Section 4.1 of [2].
> > > > > > > > >
> > > > > > > > > To avoid ambiguity, we remove the potentially misleading citation of [1] and refer readers to Appendix B.1 for the above discussion.
> > > > > > > > >
> > > > > > > > > [1] Chebotar Y, Hausman K, Lu Y, et al. Actionable Models: Unsupervised Offline Reinforcement Learning of Robotic Skills. ICML 2021.
> > > > > > > > >
> > > > > > > > > [2] Eysenbach, Benjamin, et al. Rewriting history with inverse RL: Hindsight inference for policy improvement. NIPS, 2020.

---

> > > > > > > > > > ### Comment · Reviewer_i8oK · 2021-11-23
> > > > > > > > > > **Concerns have been addressed.**
> > > > > > > > > >
> > > > > > > > > > Thanks, these revisions address my concerns in this thread.

---

> > > > > ### Author Response · Authors · 2021-11-19
> > > > > **Further clarification on your questions (part 1/2)**
> > > > >
> > > > > Thanks for your timely comments. We provide further clarification on your remaining questions.
> > > > >
> > > > > **Q1**: Could the authors also add the objective functions for GoalBCQ and GoalCQL? (I want to make sure I understand where the goals are being sampled from.)
> > > > >
> > > > > **A1**: Thanks for your suggestions. We have updated the objective functions for Goal BCQ and Goal CQL in the revision. **Since we consider the offline goal-conditioned RL setting, all the goals for training are sampled from the offline dataset**. The offline dataset is collected by a goal-conditioned policy, and each trajectory is associated with a goal given by the environment when collecting the dataset.
> > > > >
> > > > >  * Goal BCQ: goal-conditioned Batch-Constrained Q-Learning [1], which we use the official code base and concatenate observations, desired goals and achieved goals as states. The objective function of Goal BCQ is as follows:
> > > > >     $$J\_{BCQ}(\xi)=\mathbb{E}\_{(s\_t,g)\sim D, a\sim G\_w (s\_t,g)} \[Q(s\_t, a + \xi( s\_t, a, g, \Phi),g)\],$$
> > > > > where $\xi(s\_t, a ,g, \Phi)$ is the goal-conditioned perturbation model, which outputs an adjustment to an action $a$ within the range $[-\Phi,\Phi]$. $G\_w$ is the VAE fitted on the behavior policy of the offline dataset. The policy is optimized through optimizing $\xi$. The $Q$ function is learned using the Clipped Double Q-learning in [1], and the only difference is that the $Q$ function also includes goals $g$ as input.
> > > > >
> > > > > * Goal CQL: goal-conditioned Conservative Q-Learning [2], which we also use the official code base and concatenate observations, desired goals and achieved goals as states. The objective function of Goal CQL is as follows:
> > > > >     $$J\_{CQL}(\pi)=\mathbb{E}\_{(s\_t,g)\sim D, a\sim \pi(\cdot|s\_t,g)}\[Q(s\_t,a,g)-\log\pi(a|s\_t,g)\],$$
> > > > > which jointly optimizes estimated return and policy entropy. The Q function of CQL is learned by minimizing the following loss:
> > > > >     $$L\_{CQL} = \alpha \mathbb{E}\_{(s\_t,g)\sim D} \[\log\sum\_a \exp(Q(s\_t,a,g)) -\mathbb{E}\_{a\sim {\pi\_b(\cdot|s\_t,g)}}\[Q(s\_t,a,g)\] \]$$
> > > > >     $$\quad\quad\quad+ \frac{1}{2} \mathbb{E}\_{(s\_t,a\_t, s\_{t+1},g)\sim D} \[(Q(s\_t,a\_t,g) - \mathcal{B}^{\pi}Q(s\_t,a\_t,g))^2 \],$$
> > > > > where $\pi_b$ is the behavior policy of the offline dataset, which can be learned through imitation learning. $\mathcal{B}^{\pi}$ is the Bellman operator.
> > > > >
> > > > > [1] Fujimoto S, Meger D, Precup D. Off-policy deep reinforcement learning without exploration. ICML, 2019.
> > > > >
> > > > > [2] Kumar A, Zhou A, Tucker G, et al. Conservative Q-learning for offline reinforcement learning. NIPS, 2020.
> > > > >
> > > > >
> > > > > **Q2**: I didn't understand this explanation (e.g., what does the policy being random have to do with this? where does the theorem assume that $J\_{surr}\leq 0$?)
> > > > >
> > > > > **A2**: Theorem 1 assumes $\pi$ to be a random discrete policy which selects actions with non-zero probability. This assumption prevents the probability of $\pi(a|s,g)$ to be zero and guarantees $\log\pi(a|s,g)$ to exist. Based on the assumption, we have $\log\pi(a|s,g)\leq 0$ for all $a, s, g$. Given that $\gamma>0$ and $1\[\phi(s)=g\]\geq 0$ for all $s$, we have
> > > > > $$
> > > > >     J\_{surr}(\pi)=\frac{1}{T}\mathbb{E}\_{g\sim p(g),\tau\sim\pi\_b(\cdot|g)}\[\sum\_{t=0}^T\log\pi(a\_t|s\_t,g)\sum\_{i=t}^T\gamma^i\cdot 1\[\phi(s\_i)=g\]\]\leq 0.
> > > > > $$
> > > > > The same assumption is adopted in Appendix B.1 of GCSL [3].
> > > > >
> > > > > [3] Ghosh D, Gupta A, Reddy A, et al. Learning to reach goals via iterated supervised learning. ICLR 2021.

---

> > > ### Comment · Reviewer_i8oK · 2021-11-20
> > > **Discussion of propositions**
> > >
> > > Thanks for clarifying Proposition 1.
> > >
> > > Proposition 2: While I haven't checked the proof carefully[+], the *math* seems correct but the *language* used to describe it seems incorrect. The proposition seems to say that the new policy will get higher returns than the old policy, but only if you command different goals for the two policies. Thus, this seems like an apples-to-oranges comparison.
> > >
> > > I would highly recommend revising the paper so that both Propositions are stated precisely and unambiguously.
> > >
> > > [+] Looking at the proofs again, it seems like they assume that goals are discrete (otherwise, the probability of reaching the goal exactly is 0). If this is the case, I would recommend revising Section 2 to say that the analysis only applies to MDPs with discrete states/actions.
> > >
> > > [Edited on Nov 22 to fix formatting]

---

> > > > ### Author Response · Authors · 2021-11-22
> > > > **RE: Discussion of propositions**
> > > >
> > > > Thank you for the feedback!
> > > >
> > > > **Q1:**
> > > > Proposition 2: While I haven't checked the proof carefully[+], the math seems correct but the language used to describe it seems incorrect. The proposition seems to say that the new policy will get higher returns than the old policy, but only if you command different goals for the two policies. Thus, this seems like an apples-to-oranges comparison.
> > > >
> > > > I would highly recommend revising the paper so that both Propositions are stated precisely and unambiguously.
> > > >
> > > > **A1:**
> > > > Thanks for your suggestions. We previously assumed the $\phi(s_i)$ has the same distribution as $g$ ('perfectly mapped') and compared the expectation on the distribution, and thus Proposition 2 does not have the problem as the reviewer mentioned.
> > > >
> > > > Fortunately, inspired by the reviewer's question, we make efforts to derive a stronger result, which guarantees $V^{\pi\_{relabel}}(s_t,g)\geq V^{\pi\_b}(s\_t,g), \forall s\_t, g$. The updated Proposition 2 and its proof can be found in the main paper and Appendix B.4. We also refine the math language for improving clarity.
> > > >
> > > > We thank the reviewer for valuable discussions that make our work more theoretically grounded. We would appreciate your further reply that lets us know if we have addressed your concerns and if there is anything else we can do to improve our paper.
> > > >
> > > >
> > > > **Q2:**
> > > > [+] Looking at the proofs again, it seems like they assume that goals are discrete (otherwise, the probability of reaching the goal exactly is 0). If this is the case, I would recommend revising Section 2 to say that the analysis only applies to MDPs with discrete states/actions.
> > > >
> > > > **A2:**
> > > > Thanks for the suggestion! We have revised the sentence in Section 2 to show that we consider MDPs with discrete states/goals for theoretical analysis. For Theorem 1, we require discrete policy as an additional assumption. The same assumption is adopted by GCSL [1].
> > > >
> > > > [1] Ghosh D, Gupta A, Reddy A, et al. Learning to reach goals via iterated supervised learning. ICLR, 2021.

---

> > > > > ### Comment · Reviewer_i8oK · 2021-11-22
> > > > > **Discussion**
> > > > >
> > > > > > Discrete state and action spaces.
> > > > >
> > > > > I would recommend stating this when the state and action spaces are defined in the first paragraph of Section 2. Readers are liable to miss this important assumption if it is introduced many sentences after the state and action spaces are defined.
> > > > >
> > > > > > "perfectly mapped"
> > > > >
> > > > > The "perfectly mapped" assumption in Proposition 2 is different from the issue I am concerned about. The definition of this assumption in Proposition 2 says that every state can be converted into a goal. I think the assumption that we need is something about the *distribution* over the goals (i.e., an assumption on $p(\phi(s))$, rather than on $\phi(s)$). This distribution may have to be conditioned on the current state to make this assumption work.

---

> > > > > > ### Author Response · Authors · 2021-11-23
> > > > > > **RE: Discussion**
> > > > > >
> > > > > > **Q1**: I would recommend stating this when the state and action spaces are defined in the first paragraph of Section 2. Readers are liable to miss this important assumption if it is introduced many sentences after the state and action spaces are defined.
> > > > > >
> > > > > > **A1**: Thanks for your suggestions. We have added this assumption in the first paragraph of Section 2.
> > > > > >
> > > > > > **Q2**: The "perfectly mapped" assumption in Proposition 2 is different from the issue I am concerned about. The definition of this assumption in Proposition 2 says that every state can be converted into a goal. I think the assumption that we need is something about the distribution over the goals (i.e., an assumption on $p(\phi(s))$, rather than on $\phi(s)$). This distribution may have to be conditioned on the current state to make this assumption work.
> > > > > >
> > > > > > **A2**: We agree with the reviewer that some assumptions about the distribution over the goals can make the original Proposition 2 more meaningful. Inspired by the reviewer's question, we derive a new Proposition 2 in the revision, which provides a stronger result. We believe this stronger proposition can also address your concerns, as it holds for any $s\_t$ and $g$ in the dataset and does not require assumptions on the goal distribution.

---

> > ### Comment · Reviewer_i8oK · 2021-11-19
> > **Is Theorem 1 useful?**
> >
> > I wanted to continue the discussion of Theorem 1:
> >
> > > A4: Theorem 1 is useful [because] $J_\text{surr}$ has the same gradient as the actual objective $J$.
> >
> > I'm not sure that this makes Theorem 1 useful. This is a good argument for why $J_\text{surr}$ is useful (and should be emphasized more in the text!). But I'm not sure if the lower-bound relationship between these surrogate gradient estimators is useful. The root issue is that the *value* of the surrogate function doesn't seem to matter, as long as its *gradient* points in the right direction. If we want to claim that one surrogate function is better than another, we need to argue that it provides better *gradients*, right?

---

> > > ### Comment · Reviewer_fo1Z · 2021-11-20
> > > **Also interested in this answer.**
> > >
> > > I also realize that optimizing lower bound estimators of $J_{surr}(\pi)$ makes sense if optimizing such an estimator results in better values of the original objective $J(\pi)$. So, does the WGCSL objective lower bounds the true objective $J(\pi)$ too?

---

> > > > ### Author Response · Authors · 2021-11-21
> > > > **Further discussion on Theorem 1**
> > > >
> > > > Thanks for your replies. We provide further explanation of your concerns.
> > > >
> > > > **Q1**: I wanted to continue the discussion of Theorem 1:
> > > > "A4: Theorem 1 is useful [because]
> > > > $J\_{surr}$ has the same gradient as the actual objective J".
> > > > I'm not sure that this makes Theorem 1 useful. This is a good argument for why $J\_{surr}$ is useful (and should be emphasized more in the text!). But I'm not sure if the lower-bound relationship between these surrogate gradient estimators is useful. The root issue is that the value of the surrogate function doesn't seem to matter, as long as its gradient points in the right direction. If we want to claim that one surrogate function is better than another, we need to argue that it provides better gradients, right?
> > > >
> > > > **A1**: The use of Theorem 1 is to bridge the gap between $J$ and $J\_{WGCSL}$ via $J\_{surr}$, and to show that $J\_{WGCSL}$ is a tighter lower bound of $J\_{surr}$ than $J\_{GCSL}$. Given that $J\_{surr}$ has the same gradient on $\pi\_b$ with $J$ and we implicitly constraint $\pi$ to be close to $\pi\_b$, $J$ is expected to have improved if $J\_{surr}$ is improved. In addition, inspired by the reviewer's comments, we further found that $J\_{surr}$ is a lower bound of $\log J$ and $J\_{WGCSL}$ is a tighter bound of $\log J$ than $J\_{GCSL}$. The detailed proof is in Appendix B.1 Remark, which bridges the gap between $\log J$ and $J\_{surr}$. Practically, optimizing over a tighter lower bound is more likely to have better performance ([1], [2], and [3]). We agree with the reviewer that Theorem 1 might provide no strict guarantee that WGCSL can achieve larger $J$ than GCSL, but it provides some theoretical insights about why WGCSL has strong performance empirically.
> > > >
> > > > Importantly, we would also like to highlight that **Proposition 1 has already shown that WGCSL can learn a uniformly as good as or better policy $\tilde\pi$ than GCSL, where GCSL is essentially imitating the policy $\pi\_{relabel}$**.
> > > >
> > > > [1] Burda, Yuri, Roger Grosse, and Ruslan Salakhutdinov. "Importance weighted autoencoders." arXiv preprint arXiv:1509.00519 (2015).
> > > >
> > > > [2] Lemire, Daniel. "Faster retrieval with a two-pass dynamic-time-warping lower bound." Pattern recognition 42.9 (2009): 2169-2180.
> > > >
> > > > [3] Todros, Koby, and Joseph Tabrikian. "A new lower bound based on weighted Fourier transform of the likelihood ratio function." 2008 5th IEEE Sensor Array and Multichannel Signal Processing Workshop. IEEE, 2008.
> > > >
> > > > **Q2**: I also realize that optimizing lower bound estimators of $J_{surr}(\pi)$ makes sense if optimizing such an estimator results in better values of the original objective $J(\pi)$. So, does the WGCSL objective lower bounds the true objective $J(\pi)$ too?
> > > >
> > > > **A2**: Both $J\_{WGCSL}$ and $J\_{GCSL}$ are indeed lower bounds of $J$. However, we find that these two lower bounds are trivial, as $J\geq 0$ and $J\_{WGCSL}\leq 0, J\_{GCSL}\leq 0$ hold for all $\pi$. Therefore, we resort to discussing the connection between $J\_{surr}$ and $J\_{WGCSL}$ ($J\_{GCSL}$), which can provide more theoretical insights.
> > > >
> > > > Fortunately, both reviewers' comments inspire us to delve into the direct connection between $J$ and $J\_{WGCSL}$ ($J\_{GCSL}$). Based on Theorem 1, we take one step further and find more convincing theoretical results: if we bound the KL divergence between $\pi\_b$ and $\pi$ (which is a natural constraint for offline RL), $J\_{WGCSL}$ is a tighter bound of $\log J$ than $J\_{GCSL}$. The detailed proof is in Appendix B.1 Remark, which bridges the gap between $\log J$ and $J\_{surr}$. We appreciate two reviewers for valuable discussions on Theorem 1 that make our work more theoretically grounded.

---

> > > > > ### Comment · Reviewer_fo1Z · 2021-11-22
> > > > > **Thanks!**
> > > > >
> > > > > This is a pretty nice connection and I would urge the authors to highlight this in the main paper too. Overall, I feel that this paper can be accepted at the conference :)

---

> > > > > > ### Author Response · Authors · 2021-11-23
> > > > > > **Thanks for your positive feedback!**
> > > > > >
> > > > > > Thank you very much for engaging in the discussion! As the reviewer suggested, we have highlighted this new connection in the main paper right after Theorem 1 in the revision. We appreciate your comments that help us improve our paper.

---

> > > > > ### Comment · Reviewer_i8oK · 2021-11-22
> > > > > **Discussion of lower bounds**
> > > > >
> > > > > Thanks to the authors for explaining this new result. The new result seems more useful (and not misleading), so perhaps it should go in the main paper in place of Theorem 1.
> > > > >
> > > > > While I believe that Proposition 1 is correct, I don't find it convincing because it's analogous to saying the following: given a policy, if you take an (exact) gradient step then you'll get a better policy. [I think this analogy is exact if gradients are performed on policy logits.] What's missing is some notion of estimation error of that advantage function (the gradient in the analogy above).

---

> > > > > > ### Author Response · Authors · 2021-11-23
> > > > > > **RE: Discussion of lower bounds**
> > > > > >
> > > > > > Thanks for your positive feedback. As the reviewer suggested, we have highlighted this new connection in the main paper right after Theorem 1.
> > > > > >
> > > > > > **A1**: Thanks for your insightful question! We agree with the reviewer that it is important to consider the value estimation error on the policy improvement. Theoretical analysis on policy improvement with value estimation error is challenging, and many previous policy optimization algorithms (e.g., TRPO [1]) often assume access to exact value function in the analysis.
> > > > > >
> > > > > > To make our paper more theoretically grounded, we provide a new analysis in Appendix B.6 to derive the lower bound when considering the value estimation error. The proof is based on MARWIL [2] combined with value estimation error. We hope this theoretical analysis can address your concerns.
> > > > > >
> > > > > > [1] Schulman J, Levine S, Abbeel P, et al. Trust region policy optimization. ICML, 2015.
> > > > > >
> > > > > > [2] Wang Q, Xiong J, Han L, et al. Exponentially weighted imitation learning for batched historical data. NIPS, 2018.

---

> > > > > > > ### Comment · Reviewer_i8oK · 2021-11-23
> > > > > > > **Elaborate on this new results**
> > > > > > >
> > > > > > > Would the authors mind elaborating on this new result? I didn't understand how $\pi_\theta$ is learned from $A_\psi$.

---

> > > > > > > > ### Author Response · Authors · 2021-11-25
> > > > > > > > **Explanation on the new result**
> > > > > > > >
> > > > > > > > Thanks for the feedback. The new proposition shows the lower bound of the policy improvement over the behavior policy $\pi$ (in WGCSL, $\pi$ is corresponding to the behavior policy of relabeled dataset), considering a parametric policy $\pi\_\theta$ learned from an estimated advantage function $A\_\psi$.
> > > > > > > >
> > > > > > > > For simplicity, we omit goals $g$ in the following analysis, which can be considered as new states combining states and goals.
> > > > > > > > We consider exponential advantage weight in WGCSL.
> > > > > > > > If we know the exact advantage $A$, we can imitate a policy $\tilde\pi$ with the exponential weight in our setting: $\tilde{\pi}(a|s)=\pi(a|s)\exp(A^{\pi}(s,a))$.
> > > > > > > > Practically, we first learn a parametric $A\_\psi$ to approximate $A$, and then reweight the dataset with $\exp(A^{\pi}\_{\psi}(s,a))$. We use supervised learning on the reweighted dataset to learn the parametric policy $\pi\_\theta$.
> > > > > > > >
> > > > > > > > From the new proposition in Appendix B.6, we can observe that the lower bound of the policy improvement has three terms, and we explain each term with varying on $\pi\_\theta$.
> > > > > > > >
> > > > > > > > (1) $-\frac{\sqrt{2}}{1-\gamma} \delta\_1^{\frac{1}{2}} (M^{\pi\_\theta}+\xi\_A)$: as shown in the proposition, $M^{\pi\_\theta}$ is upper bounded by a constant that does not depend on $\pi\_\theta$. $\delta\_1$ measures the distance between $\pi\_\theta$ and $\tilde\pi$. $\xi\_A$ measures the distance between $A$ and $A\_\psi$, so that $J(\pi\_\theta)-J(\pi)$ will become larger if $A$ is more accurately estimated by $A\_\psi$. Moreover, since $\pi\_\theta$ can match the exponential weighted dataset through supervised learning with sufficient data, a better estimation $A\_\psi$ can lead $\pi\_\theta$ to be closer to $\tilde\pi$, making $\delta\_1$ smaller and thus enlarging the policy improvement.
> > > > > > > >
> > > > > > > > (2) $\frac{1}{(1-\gamma)\beta}\delta\_2$: $\delta\_2$ measures the distance between $\tilde\pi$ and $\pi$, which does not depend on $\pi\_\theta$.
> > > > > > > >
> > > > > > > > (3) $- \frac{\sqrt{2}\gamma\epsilon\_{\pi}^{\tilde{\pi}}}{(1-\gamma)^2}\delta\_2^{\frac{1}{2}}$: again, both $\epsilon\_{\pi}^{\tilde{\pi}}$ and $\delta\_2$ does not depend on $\pi\_\theta$.
> > > > > > > >
> > > > > > > > Combining (1)(2)(3), if we have a better estimation $A\_\psi$ of $A$, then $\pi\_\theta$ can have more policy improvement over $\pi$.
> > > > > > > >
> > > > > > > > [1] Wang Q, Xiong J, Han L, et al. Exponentially weighted imitation learning for batched historical data. NIPS, 2018.
> > > > > > > >
> > > > > > > >
> > > > > > > > Errata in the proof of the new proposition in Appendix B.6:
> > > > > > > >
> > > > > > > > 1. $\epsilon\_{\pi}^{\tilde{\pi}} = \max\_{s} |\mathbb{E}\_{a \sim \pi'} A^\pi (s,a,g)|$ should be $\epsilon\_{\pi}^{\tilde{\pi}} = \max\_{s} |\mathbb{E}\_{a \sim \tilde{\pi}} A^\pi (s,a,g)|$
> > > > > > > > 2. $\beta$ is the hyperparameter controlling the importance of advantage weighting, in WGCSL, we can assume $\beta=1$
> > > > > > > > 3. $D_{KL}^d(\pi' \| \pi)$ is defined as $D\_{KL}^d(\pi' \| \pi)=\sum\_s d(s)\sum\_a \pi'(a|s) \log \frac{\pi'(a|s)}{\pi(a|s)}$
> > > > > > > > 4. $D_{TV}^d(\pi' \| \pi)$ is defined as $D\_{TV}^d(\pi' \| \pi)=\frac{1}{2} \sum\_s d(s)\sum\_a |\pi'(a|s)-\pi(a|s)|$

---

> ### Author Response · Authors · 2021-11-28
> **Looking forward to further feedback**
>
> Dear Reviewer,
>
> Thank you for your time and efforts in reviewing our work. We appreciate your inspiring comments and discussions, which greatly help improve our work. We hope our response and clarifications have addressed your concerns. We would be grateful if you could re-evaluate our work. If you have any additional questions or comments, we would be happy to have further discussions.
>
> Best,
>
> The authors

---

> > ### Comment · Reviewer_i8oK · 2021-11-29
> > **Discussion**
> >
> > Thanks for following up; I apologize for the delayed response.
> >
> > I am happy with the empirical contributions of the paper. The experiments seem thorough and conclusive. If the paper did not include any theoretical contributions, I would feel fine accepting it. Given that the paper does include theoretical contributions, I think it's important to ensure the correctness of these contributions. I'm still a bit uneasy about a few parts of the theory;
> >
> > 1. While Theorem 1 is correct, I find it somewhat ingenuous (as previously discussed). While prior work has performed a similar slight-of-hand, I feel like it shouldn't be swept under the rug. The new result about $\log J$ is much, much better, and I would be happy to see Theorem 1 *replaced* by that result. As I am obligated to review the paper as written (not the potential camera ready version), I see this as a point of concern.
> >
> > 2. I'm not sure if the new result about $\log J$ is correct. My reasoning is that we can construct datasets where goal-conditioned behavior cloning (and GCSL) converge to a suboptimal policy. For example, let's say an agent tries to solve a task using strategy A 99% of the time and strategy B 1% of the time. Even if strategy B is more effective at solving the task, the GCBC agent will end up preferring strategy A if strategy A is effective at least some fraction of the time. The discounting used in this paper doesn't avoid the problem; I'm not sure if the advantage weighting avoids the problem.
> >
> > 3. The hunch that the new result isn't correct (#2) made me look at the proof of this result more closely (Remark on page 15). Where is goal relabeling introduced in this proof?

---

> > > ### Author Response · Authors · 2021-11-29
> > > **Response to Discussion**
> > >
> > > Thanks for your reply! We are pleased that the reviewer found our empirical contributions convincing and conclusive. We totally agree with the reviewer that it is important to ensure the correctness of theoretical results. As to the correctness of the new result about $\log J$, the reviewer may have some misunderstandings about its implication and its proof. We provide clarification as below and hope it would address the reviewer's concerns. If the reviewer has any questions about the correctness of Theorem 1, Proposition 1, Proposition 2, or the new result, we will be happy to have further discussions.
> > >
> > > **A1:** As discussed in our previous response, the original Theorem 1 is useful, because $J\_{surr}$ shares the same gradient with $J$, such that a tighter lower bound of $J\_{surr}$ can possibly facilitate the optimization over $J\_{surr}$ and hence benefit optimizing $J$. As we have claimed in previous discussions, Theorem 1 only shows a lower bound connection between $J\_{surr}$ and $J\_{WGCSL}$, but cannot guarantee that WGCSL is better than GCSL. In fact, the property that WGCSL is better than GCSL is proved by our Proposition 1.
> > >
> > > We really appreciate the reviewer for extensive discussions that motivate us to establish **more general results on $\log J$, which have been included in the revised submission**. In our humble opinion, the revision is joint effort among reviewers, authors, and public readers; and it shall be considered in the discussion period. To improve the clarity, we are willing to merge the current Theorem 1 and the emphasized new result in the submitted PDF as a more comprehensive theorem in the final version.
> > >
> > > **A2:** There might be some misunderstanding about our theory and our paper. First of all, the new result about $\log J$ does not imply the optimality of WGCSL and GCSL, as **optimizing a lower bound does not guarantee the optimal policy. Instead, we use Proposition 1 to ensure the policy improvement** with additional goal-conditioned exponential weight (GEAW) and best-advantage weight (BAW). In the example the reviewer mentioned, both GEAW and BAW contribute to the policy learning, because they will assign larger weights on those more effective trajectories (strategy B), which implicitly change the distribution of the highly-imbalanced dataset.
> > >
> > > We provide additional experimental results to verify the effectiveness of WGCSL. In our additional experiments, we use two trajectories shown in the top right subplot in Figure 2, denoted as $t\_1$ (the trajectory return is 26) and $t\_2$ (the trajectory return is 45). The mixed buffer contains 990 trajectory $t\_1$ and 10 trajectory $t\_2$, as suggested by the reviewer. We compare the final returns of different algorithms in the following table. The results demonstrate that WGCSL can learn optimal policy even if the good trajectory is very few in the offline dataset. As the reviewer pointed out, the discounting weight (DRW) is not quite helpful in this example. In contrast, BAW and GEAW significantly improve policy learning, which is the same as the results in Figure 5 of our main paper.
> > >
> > > |  Algorithms  | WGCSL  | gamma+GCSL  | GCSL | Goal BC|
> > > |  ----  | ----  |  ----  | ----  |   ----  |
> > > | Final Return  | 44.92 $\pm$ 0.12 | 29.33 $\pm$ 2.54 | 30.50 $\pm$ 2.11 | 26.33 $\pm$ 0.82 |
> > >
> > > **A3:**
> > > We have double checked the proof of the new result about $\log J$ and we are confident on its correctness. If the reviewer is aware of some mistakes in the proof, we will be happy to have further discussions.
> > >
> > > Regarding the reviewer's question, the Remark on page 15 does not require goal relabeling, because it aims at proving $J\_{surr}$ lower bounds $\log J$. Goal relabeling is used to show that WGCSL (and GCSL) provides a lower bound of $J\_{surr}$, which is proved in Theorem 1.

---

> > > > ### Comment · Reviewer_i8oK · 2021-11-30
> > > > **Discussion**
> > > >
> > > > Hi authors,
> > > >
> > > > A1 -- The response here doesn't change my mind, as it simply states information that was already discussed in other threads. (To be fair, the only way I can think I'd be convinced on this point is to see a revised version of the paper, which isn't possible at this point.)
> > > >
> > > > A2 -- Thanks for clarifying that this lower bound isn't tight, and for running the additional experiment.
> > > >
> > > > A3 -- Thanks for clarifying this point. I don't see any bugs in the proof (ignoring small typos). I'm still think there's something that I'm missing because the logical conclusion from this result is that goal-conditioned behavior cloning objective, $J_\text{GCSL}$, results in sensible learning dynamics. But, as discussed earlier, this is not the case. In fact, you can construct examples where iterating goal conditioned behavior cloning results in the policy getting worse and worse at reaching the goal.
> > > >
> > > > There might be a few ways to make sense of this:
> > > > 1. The lower bound in the Remark is very loose.
> > > > 2. The lower bound in the LHS of Theorem 1 is very loose.
> > > > 3. The lower bound in the RHS of Theorem 1 is very loose.
> > > > 4. There's a bug in one of the proofs.

---

> > > > > ### Author Response · Authors · 2021-11-30
> > > > > **Response to the Discussion**
> > > > >
> > > > > Thanks for your feedback. We are pleased that our response has  addressed some of your concerns. Here we provide further clarification.
> > > > >
> > > > >
> > > > > **Q1:** A1 -- The response here doesn't change my mind, as it simply states information that was already discussed in other threads. (To be fair, the only way I can think I'd be convinced on this point is to see a revised version of the paper, which isn't possible at this point.)
> > > > >
> > > > > **A1:** It is unfortunate that we cannot submit a revision on this point. However, we do think that the current version have properly presented Theorem 1 and highlighted the new result in the main text (page 4), if it is not clear to the reviewer. We sincerely hope you could take them into re-consideration.
> > > > >
> > > > > **Q2:** A2 -- Thanks for clarifying that this lower bound isn't tight, and for running the additional experiment.
> > > > >
> > > > > **A2:** We are glad about our response has addressed your concern and clarified potential misunderstandings.
> > > > >
> > > > > **Q3:** A3 -- Thanks for clarifying this point. I don't see any bugs in the proof (ignoring small typos). I'm still think there's something that I'm missing because the logical conclusion from this result is that goal-conditioned behavior cloning objective,
> > > > > $J\_{GCSL}$, results in sensible learning dynamics. But, as discussed earlier, this is not the case. In fact, you can construct examples where iterating goal conditioned behavior cloning results in the policy getting worse and worse at reaching the goal.
> > > > >
> > > > > **A3**: Thank you for the feedback. However, we have some confusion about the relevance between the example mentioned and the correctness of Theorem 1. In fact, if the policy is commonly randomly initialized, we do not think it will get worse and worse during the training process. Take the unbalanced dataset 99\% A and 1\% B in the previous discussion as an example. If we train the policy with GCSL (or GCBC), the policy will gradually converge to the behavior policy of the relabeled dataset (or the dataset). While the behavior policy is not the optimal policy, the performance of the learning policy will gradually improve until convergence while training. The difference between GCSL and WGCSL is that: WGCSL will reweight the dataset distribution and hence converge to a better policy which is more effective at reaching the goal, as is shown in our additional experiment in the previous response and all our results in the main paper.

---

> > > ### Author Response · Authors · 2021-11-30
> > > **Thank you for your time and efforts in reviewing our work!**
> > >
> > > Thank you again for your time and efforts in engaging in the discussions, which help improve our work. We hope our responses and new theoretical results have addressed your concerns. As the discussion period ends soon, we sincerely appreciate it if you could re-evaluate our work. We will also be happy to have further discussions if you have any additional questions.

---

### Author Response · Authors · 2021-11-17
**Overall Response**

We thank all the reviewers for the constructive feedback and for finding our work a "novel method" (Reviewer i8oK), "well written" (Reviewer i8oK, FPSj) while presenting "strong empirical results" (Reviewer i8oK, FPSj, fo1Z).

In summary, our response includes the following aspects:

1. Providing additional baseline results for Actionable Models and Goal AWR.
2. Providing additional analysis, e.g., the training time comparison and the normalized scores across 10 tasks.
3. Refining our paper to provide clarification to reviewers, e.g., typos and the notion issues in our theory part.

We hope our response could address the reviewers' concerns. If you have any other questions, please post them, and we are happy to have further discussions.

---

### Decision · Program_Chairs · 2022-01-20

**Decision:**

Accept (Poster)

**Comment:**

The authors introduce a method that improves goal-conditioned supervised learning (GCSL) by iteratively re-weighting the experience by a variable that correlates with the number of steps till the desired goal. The reviewers mention that the authors focus on an important problem, their method is simple and the empirical results are significant. However, they do point several flaws of the paper, the main ones being questionable theoretical claims and the clarity of the presentation. After an extensive discussion, most reviewers agree that the paper should be accepted but I do encourage the authors to take into account the comments by the reviewers for the final version of the paper and make the theory more clear.